# Generalization of Diffusion Models Arises with a Balanced Representation Space

**Zekai Zhang**[*]  **Xiao Li**[*]  **Xiang Li**  **Lianghe Shi**  **Meng Wu**
University of Michigan

**Molei Tao**
Georgia Institute of Technology

**Qing Qu**[†]
University of Michigan

## Abstract

Diffusion models excel at generating high-quality, diverse samples, yet they risk memorizing training data when overfit to the training objective. We analyze the distinctions between memorization and generalization in diffusion models through the lens of representation learning. By investigating a two-layer ReLU denoising autoencoder (DAE), we prove that: (*i*) memorization corresponds to the model storing raw training dataset in the learned weights for encoding and decoding, yielding localized, spiky representations; whereas (*ii*) generalization arises when the model captures local data statistics, producing balanced representations. Furthermore, we validate our theoretical findings on real-world unconditional and text-to-image diffusion models, demonstrating that the same representation structures emerge in deep generative models with significant practical implications. Building on these insights, we propose a representation-based method for detecting memorization and a training-free editing technique that allows precise control via representation steering. Together, our results highlight that *learning good representations is central to novel and meaningful generative modelling*. Code is available at `https://github.com/la0ka1/diffusion-gen-from-rep`.

## 1 Introduction

Diffusion models (Ho et al., 2020; Lou et al., 2024) have rapidly emerged as the dominant class of generative models, powering state-of-the-art systems such as Stable Diffusion (Rombach et al., 2022), Flux (Labs et al., 2025), and Veo (Google, 2025). By iteratively denoising random noise, they achieve unprecedented scalability, controllability, and fidelity. However, their empirical success raises a fundamental question: in principle, the standard training objective (e.g., denoising score matching) admits a closed-form solution that merely memorizes training examples (Yi et al., 2023); in practice, however, real-world models consistently produce novel and diverse outputs (Zhang et al., 2024; Kadkhodaie et al., 2024a). This distinct mismatch between theoretical expectation and observed behavior poses a critical gap in our *understanding of diffusion model generalization*, with direct implications for privacy, interpretability, and trustworthy deployment (Somepalli et al., 2023a).

Addressing this question has drawn increasing attention in the machine learning community (Zhang et al., 2024; Li et al., 2024b; Wang et al., 2024a; Kadkhodaie et al., 2024a; Gu et al., 2025; Bonnaire et al., 2025; Zhang et al., 2025b; Bertrand et al., 2025; Zhang et al., 2025a), yet existing explanations remain far from satisfactory. Early works based on random feature models (Li et al., 2023; George et al., 2025) provide useful insights but necessarily oversimplify model architectures. Analyses of linear models on Gaussian mixtures (Li et al., 2024b; Wang et al., 2024a; Wang, 2025) shed light on generalization but cannot capture memorization. Another line of research explores inductive biases by constructing handcrafted closed-form solutions from empirical data to approximate U-Net performance (Kamb and Ganguli, 2025; Niedoba et al., 2025; Lukoianov et al., 2025; Floros et al., 2025), attributing success to principles such as locality and equivariance. While these advances are valuable, the findings remain fragmented and phenomenological, and a more unified account of how diffusion models both memorize and generalize is still lacking (see Appendix A for a more detailed discussion of related work).

---

[*]Equally contributing, Email: `zzekai@umich.edu`
[†]Corresponding author. Email: `qingqu@umich.edu`

Figure 1: **Diffusion models generalize while learning benign internal representations.** Activations from intermediate network layers form a *representation space*, within which distinct patterns emerge: memorized samples produce spiky representations that make them detectable, whereas novel generations yield balanced, information-rich representations that support controllable generation via representation steering.

To address these challenges, we develop a unified mathematical framework based on a theoretical analysis of a nonlinear two-layer ReLU denoising autoencoder (DAE) (Vincent, 2011). This framework not only unifies the characterization of memorization and generalization, but also bridges distribution learning with representation learning, offering profound practical implications. Specifically: (*i*) **Memorization.** We prove that when empirical samples are locally sparse, the network weights memorize and store individual training examples, leading to overfitting and hence memorization. (*ii*) **Generalization.** Conversely, when the empirical data are locally abundant, the weights effectively capture local data statistics, enabling the model to generate novel in-distribution samples.

Crucially, our work provides a unique **representation-centric** perspective on generalization (Tian, 2025), highlighting the pivotal role of bottleneck activations in DAE networks. This view is motivated by recent empirical evidence on the duality between distribution learning and representation learning in diffusion models (Li et al., 2025c; Xiang et al., 2025; Tinaz et al., 2025): they inherently learn informative features for downstream tasks (Kwon et al., 2023; Chen et al., 2025b), and representation alignment regularization has been shown to accelerate training (Yu et al., 2025). Our theory makes this connection explicit: memorized samples are encoded as spiky activations concentrated on a few neurons, whereas generalized samples yield balanced representations that reflect the underlying distribution. These contrasting modes of representation learning manifest in distinct generation behaviors in terms of memorization or generalization, which we comprehensively validated across a range of models, including EDM (Karras et al., 2022), Diffusion Transformers (DiT) (Peebles and Xie, 2023), and Stable Diffusion v1.4 (Rombach et al., 2022) (SD1.4).

Moreover, our findings show that the representation space is not a byproduct but a crucial and controllable factor for generation. Specifically, we demonstrate two practical implications: (*i*) **Memorization detection.** Leveraging the spikiness of representations identified by our theory as a signature of memorization, we develop a theory-driven detector that achieves highly accurate and efficient performance in a prompt-free manner. (*ii*) **Model steering.** We propose an effective steering method based on additions in the representation space and reveal distinct behaviors between memorization and generalization: memorized samples are difficult to steer, whereas generalized samples are highly steerable owing to their balanced, semantically rich representations. Together, these applications illustrate the far-reaching implications of our representation-centric analysis for the privacy, interpretability, and controllability of diffusion models.

**Summary of contributions.** Our main contributions are as follows:

- **Unified framework in a nonlinear ReLU setting.** We analyze the optimal solutions of a two-layer nonlinear ReLU DAE under different empirical data sizes, providing a unified characterization of memorization and generalization that goes beyond prior random-feature or linear model analyses.

- **A representation-centric understanding of generalization.** We establish a rigorous connection between representation structures and generalization, identifying distinct patterns that separate memorization from generalization and validating these insights across diverse model settings.

- **Theory-inspired tools for memorization detection and model steering.** Building on our analysis, we propose simple yet effective methods for memorization detection and representation-space steering, revealing distinct behaviors of generalized versus memorized samples.

## 2    PROBLEM SETUP

In this section, we first introduce the basics of diffusion models, and then describe our problem setup for theoretical studies in Section 3.

### 2.1    A DENOISING PERSPECTIVE OF DIFFUSION MODELS

**Basics of diffusion models.** Diffusion models comprise two processes: (i) a forward noising process and (ii) a reverse denoising/sampling process. The forward process progressively corrupts a clean sample $\boldsymbol{x}_0$ via $\boldsymbol{x}_t = \boldsymbol{x}_0 + \sigma_t \boldsymbol{\epsilon}$ with $\boldsymbol{\epsilon} \sim \mathcal{N}(\boldsymbol{0}, \boldsymbol{I})$, while the reverse process (e.g., DDIM (Song et al., 2021a)) removes noise to generate data:

$$\boldsymbol{x}_{t-1} = \boldsymbol{x}_t - (\sigma_t - \sigma_{t-1})\,\sigma_t \nabla \log p_t(\boldsymbol{x}_t), \tag{1}$$

where $\nabla \log p_t(\boldsymbol{x}_t)$ is the score function of the marginal distribution of the noisy sample $\boldsymbol{x}_t$ at time $t$. To estimate $\nabla \log p_t(\boldsymbol{x}_t)$, we use a denoising autoencoder (DAE) $\boldsymbol{f}_{\boldsymbol{\theta}}(\boldsymbol{x}_t)$ (Karras et al., 2022; Li and He, 2025) that predicts $\boldsymbol{x}_0$ from $\boldsymbol{x}_t$, so that

$$\nabla \log p_t(\boldsymbol{x}_t) = (\boldsymbol{x}_t - \boldsymbol{f}_{\mathrm{gt}}(\boldsymbol{x}_t))/\sigma_t^2 \approx (\boldsymbol{x}_t - \boldsymbol{f}_{\boldsymbol{\theta}}(\boldsymbol{x}_t))/\sigma_t^2,$$

where $\boldsymbol{f}_{\mathrm{gt}}(\boldsymbol{y}) := \mathbb{E}[\boldsymbol{x} \mid \boldsymbol{x} + \sigma_t \boldsymbol{\epsilon} = \boldsymbol{y}; \boldsymbol{x} \sim p_{\mathrm{gt}}]$ is the ground-truth denoiser via Tweedie's formula (Efron, 2011). Thus, the ideal (population) objective to learn the DAE is

$$\frac{1}{T} \sum_{t=0}^{T} \mathbb{E}_{\boldsymbol{x} \sim p_{\mathrm{gt}},\, \boldsymbol{\epsilon} \sim \mathcal{N}(\boldsymbol{0}, \boldsymbol{I})} \left[ \left\| \boldsymbol{f}_{\boldsymbol{\theta}}(\boldsymbol{x} + \sigma_t \boldsymbol{\epsilon}, t) - \boldsymbol{x} \right\|^2 \right]. \tag{2}$$

**Generalization of diffusion models.** In practice, we only have finitely many empirical samples $\boldsymbol{X} = \{\boldsymbol{x}_i\}_{i=1}^n$ with $\boldsymbol{x}_i \sim p_{\mathrm{gt}}$. Accordingly, we work with the empirical distribution $p_{\mathrm{emp}} = \frac{1}{n} \sum_{i=1}^n \delta(\boldsymbol{x} - \boldsymbol{x}_i)$, and Equation (2) reduces to its empirical counterpart. Minimizing this empirical loss leads to the nonparametric *empirical denoiser* $\boldsymbol{f}_{\mathrm{emp}}$ (Gu et al., 2025), which maps a noisy input towards the nearest training samples:

$$\boldsymbol{f}_{\mathrm{emp}}(\boldsymbol{y}) = \mathbb{E}[\boldsymbol{x} \mid \boldsymbol{x} + \sigma_t \boldsymbol{\epsilon} = \boldsymbol{y}; \boldsymbol{x} \sim p_{\mathrm{emp}}] = \frac{\sum_{i=1}^n \mathcal{N}(\boldsymbol{y}; \boldsymbol{x}_i, \sigma_t^2 \boldsymbol{I}) \, \boldsymbol{x}_i}{\sum_{i=1}^n \mathcal{N}(\boldsymbol{y}; \boldsymbol{x}_i, \sigma_t^2 \boldsymbol{I})}. \tag{3}$$

Sampling with $\boldsymbol{f}_{\mathrm{emp}}$ can provably reproduce training samples (Zhang et al., 2024; Baptista et al., 2025). In practice, however, this empirical loss is minimized by taking the gradient descent over a parameterized neural network, which does not always overfit; instead, it can approximate the population denoiser $\boldsymbol{f}_{\mathrm{gt}}$ (Niedoba et al., 2025). In this paper, we aim to understand when a parameterized network overfits (learns $\boldsymbol{f}_{\mathrm{emp}}$) versus generalizes (learns $\boldsymbol{f}_{\mathrm{gt}}$).

### 2.2    OUR THEORETICAL FRAMEWORK

**Data assumptions.** We assume a $K$-component mixture of Gaussians (MoG) for the data distribution:

$$\boldsymbol{x} \sim p_{\mathrm{gt}} := \sum_{k=1}^K \rho_k \mathcal{N}(\boldsymbol{\mu}_k, \boldsymbol{\Sigma}_k), \qquad \sum_{k=1}^K \rho_k = 1, \tag{4}$$

which is a standard approximation to data manifolds used in recent theoretical studies (Wang et al., 2024a; Zhang et al., 2024; Li et al., 2025c; Cui and Zdeborová, 2023; Gatmiry et al., 2025; Biroli et al., 2024; Kamkari et al., 2024; Buchanan et al., 2025; Li et al., 2025c;b).

**Model parameterization and training loss.** Following Vincent (2011); Chen et al. (2023); Zeno et al. (2023); Cui et al. (2025), we parameterize the DAE by a two-layer ReLU network:

$$\boldsymbol{f}_{\boldsymbol{W}_2, \boldsymbol{W}_1}(\boldsymbol{x}) = \boldsymbol{W}_2 \boldsymbol{h}(\boldsymbol{x}) = \boldsymbol{W}_2 [\boldsymbol{W}_1^\top \boldsymbol{x}]_+, \tag{5}$$

with $\boldsymbol{W}_1, \boldsymbol{W}_2 \in \mathbb{R}^{d \times p}$, $[\cdot]_+$ denoting ReLU and $\boldsymbol{h}(\cdot)$ *stands for the representation*. Training and sampling can be viewed as operating with a collection of DAEs across multiple noise levels. Following prior work (Li et al., 2024b; Zeno et al., 2025; Zhang and Pilanci, 2024; Han et al., 2025), we begin with a fixed noise level $\sigma$. The $\ell_2$-regularized training objective is

$$\min_{\boldsymbol{W}_2, \boldsymbol{W}_1} \; \mathcal{L}_{\boldsymbol{X}}(\boldsymbol{W}_2, \boldsymbol{W}_1) = \frac{1}{n} \sum_{i=1}^n \mathbb{E}_{\boldsymbol{\epsilon} \sim \mathcal{N}(\boldsymbol{0}, \boldsymbol{I})} \left[ \left\| \boldsymbol{f}_{\boldsymbol{W}_2, \boldsymbol{W}_1}(\boldsymbol{x}_i + \sigma \boldsymbol{\epsilon}) - \boldsymbol{x}_i \right\|_2^2 \right] + \lambda \sum_{l=1}^2 \|\boldsymbol{W}_l\|_F^2. \tag{6}$$

Figure 2: **Sampling with Mem./Gen. ReLU DAEs.** *Left*: sampling with a set of memorized ReLU DAE produces duplications of training images. *Right*: sampling with generalized DAEs produces novel images. Details for training and sampling are provided in Appendix C.1 and single-step denoising results are shown in Appendix B.2

Figure 2 illustrates training and sampling across multiple noise levels under this setting; we revisit the effect of different noise levels after Corollary 3.3.

We adopt (5) as a minimal, tractable model to analyze memorization and generalization. Recent works (Lukoianov et al., 2025; Li et al., 2024b; Wang and Vastola, 2024) imply that real diffusion models exhibit approximate piecewise linearity; our ReLU model shares this structure and can be viewed as a local approximation of such networks. We verify this connection via an SVD analysis of denoiser Jacobians (Kadkhodaie et al., 2024a; Achilli et al., 2024) for EDM, SD1.4, and ReLU DAE in Appendix B.3: around generalized samples, the Jacobian reflects local data statistics as in Cor. 3.3, whereas around memorized samples it becomes noticeably low-rank and is dominated by the corresponding data vector, consistent with Cor. 3.2.

## 3  MAIN THEOREMS

Building on the setup in Section 2, this section presents our main theoretical results for a two-layer nonlinear DAE with the ReLU activation, complemented by experiments on state-of-the-art diffusion models. By characterizing the optimal solutions of the training loss, we establish:

---

**Three Learning Regimes of Training Diffusion Models**

- **Memorization Regime (Section 3.1):** In over-parameterized models trained on locally sparse data, memorization arises when network weights store individual training samples, leading to overfitting and producing distinctively spiky representations.

- **Generalization Regime (Section 3.2):** In contrast, when the model is under-parameterized and the data are locally abundant, the weights capture underlying data statistics, enabling novel sample generation and yielding balanced, semantically rich representations.

- **Hybrid Regime (Section 3.3):** Imbalanced real-world data leads to a hybrid regime where models generalize on abundant clusters while memorizing scarce ones. Consequently, the representations can help identify an input's region and detect its memorized samples.

---

To substantiate the above results, we first establish a general theorem characterizing the local minimizers of the training loss (6) for the DAE networks. This theorem then specializes to individually address the memorization and generalization regimes. To simplify the nonlinear DAE problem and obtain a more interpretable characterization, we adopt the following separability notion. It is designed to match bias-free linear layers (as in our ReLU DAE), where cluster structure is naturally captured by within-cluster concentration and angular separation of cluster means; the definition can be extended to standard hyperplane separability by allowing affine (biased) layers.

**Definition 3.1** (($\alpha, \beta$)-Separability of Training Data). *Suppose the training dataset $\boldsymbol{X}$ can be partitioned into $M$ clusters $\boldsymbol{X} = [\boldsymbol{X}_1, \dots, \boldsymbol{X}_M]$, where $\boldsymbol{X}_k = [\boldsymbol{x}_{k,1}, \dots, \boldsymbol{x}_{k,n_k}] \subseteq \mathbb{R}^d$ has mean $\bar{\boldsymbol{x}}_k := \frac{1}{n_k} \sum_{j=1}^{n_k} \boldsymbol{x}_{k,j}$. We say the dataset is $(\alpha, \beta)$-separable if*

$$\text{for all } k, j: \quad \frac{\|\boldsymbol{x}_{k,j} - \bar{\boldsymbol{x}}_k\|_2}{\|\bar{\boldsymbol{x}}_k\|_2} \leq \alpha, \qquad \text{and for all } k \neq \ell: \quad \frac{\langle \bar{\boldsymbol{x}}_k, \bar{\boldsymbol{x}}_\ell \rangle}{\|\bar{\boldsymbol{x}}_k\|_2 \|\bar{\boldsymbol{x}}_\ell\|_2} \leq \beta.$$

The parameters $\alpha$ and $\beta$ are not required to be universal constants. Intuitively, tight within-cluster concentration together with well-separated means yields an inter-cluster margin $\gamma$ that quantifies

negative alignment between samples from different clusters; $\gamma$ depends only on $\alpha$, $\beta$, and the norms of the training data (the explicit expression is given in Appendix D.2). Under this separability condition, we show that local minimizers of the DAE admit a block-wise structure.

**Theorem 3.1** (Block-wise Structure of Local Minimizers in the DAE Loss). *Suppose the training data $\boldsymbol{X} = [\boldsymbol{X}_1, \ldots, \boldsymbol{X}_M]$ is $(\alpha, \beta)$-separable according to Definition 3.1 with $\beta < 0$. Consider minimizing the training loss (6) for a DAE trained with a fixed noise level $\sigma \geq 0$ and weight decay $\lambda \geq 0$. Then there exists a local minimizer with a block-wise structure, where the weights satisfy $\boldsymbol{W}_2^\star = \boldsymbol{W}_1^\star$ where*

$$\boldsymbol{W}_1^\star = (\boldsymbol{W}_{\boldsymbol{X}_1} \quad \boldsymbol{W}_{\boldsymbol{X}_2} \quad \cdots \quad \boldsymbol{W}_{\boldsymbol{X}_M}) + \boldsymbol{R}(\sigma, \gamma). \tag{7}$$

*Here, $\boldsymbol{W}_{\boldsymbol{X}_k} \in \mathbb{R}^{d \times p_k}, \sum_{k=1}^M p_k = p$ denotes the block-decomposition of $\boldsymbol{W}$, $\boldsymbol{R}(\sigma, \gamma)$ is a small residual term whose Frobenius norm is bounded by $\|\boldsymbol{R}(\sigma, \gamma)\|_F^2 \leq C\left(e^{-c\gamma^2/\sigma^2}\right)$ for universal constants $C, c > 0$ and a margin $\gamma > 0$ determined by $(\alpha, \beta)$. Each block $\boldsymbol{W}_{\boldsymbol{X}_k}$ $(1 \leq k \leq M)$ is constructed from the Gram matrix $\boldsymbol{X}_k \boldsymbol{X}_k^\top = \boldsymbol{U}_k \boldsymbol{\Lambda}_k \boldsymbol{U}_k^\top$ of the $k$-th data cluster as follows:*

$$\boldsymbol{W}_{\boldsymbol{X}_k} = \boldsymbol{U}_k^{(p_k)} \left(\boldsymbol{I} + n_k \sigma^2 \left(\boldsymbol{\Lambda}_k^{(p_k)}\right)^{-1}\right)^{-\frac{1}{2}} \left(\boldsymbol{I} - n\lambda \left(\boldsymbol{\Lambda}_k^{(p_k)}\right)^{-1}\right)^{\frac{1}{2}} \boldsymbol{O}_k^\top, \tag{8}$$

*where (i) $\boldsymbol{U}_k^{(p_k)} \in \mathbb{R}^{d \times p_k}$ is the submatrix of $\boldsymbol{U}_k$ containing its top $p_k$ eigenvectors, (ii) $\boldsymbol{\Lambda}_k^{(p_k)} \in \mathbb{R}^{p_k \times p_k}$ contains the corresponding $p_k$ eigenvalues, and (iii) $\boldsymbol{O}_k \in \mathbb{R}^{p_k \times p_k}$ is an orthogonal matrix accounting for rotational symmetry. This holds under the condition $n\lambda < \min_k \lambda_{\min}(\boldsymbol{\Lambda}_k^{(p_k)})$, which ensures that the matrix square roots in (8) are well-defined.*

**Remarks.** The proof is deferred to Appendix D.2. The local minimizer (7) consists of a block-wise main term plus a residual $\boldsymbol{R}(\sigma, \gamma)$, which vanishes as $\sigma$ becomes small relative to the separation margin $\gamma$. This is consistent with the low-noise regimes that are crucial for diffusion-model sampling and representation learning (Niedoba et al., 2025; Pavlova and Wei, 2025). Empirically, we observe this block-wise structure even for relatively large $\sigma$ (Figure 3). The $(\alpha, \beta)$-separability assumption serves mainly to simplify the proof; similar conclusions hold more generally (see Appendix B.1). Finally, the optimal solution is not tied to a specific block order, since $\boldsymbol{f}_{\boldsymbol{W}_2, \boldsymbol{W}_1}$ is invariant to arbitrary column permutations of the weight matrices $(\boldsymbol{W}_1, \boldsymbol{W}_2)$.

For the remainder of this section, we specialize the result to the memorization (Section 3.1) and generalization (Section 3.2) regimes by varying the training-set size. For clarity, we omit the residual term $\boldsymbol{R}(\sigma, \gamma)$ and focus on the block-wise leading component of the optimal solution.

## 3.1 CASE 1: MEMORIZATION WITH OVERPARAMETERIZATION

First, we consider the overparameterized setting where the model parameters are larger than the number of training samples $p \geq n$. In this "sample sparse" regime, each training sample can be treated as an individual cluster that is sufficiently separated from each other, where $\alpha_1 = 0$ and $\beta_1$ can be set to $\max_{i,j}\langle \boldsymbol{x}_i, \boldsymbol{x}_j \rangle$. Based on this setup, Theorem 3.1 can be reduced to the following.

**Corollary 3.2** (Memorization in Overparameterized DAEs). *Under the problem setup of Theorem 3.1, consider training data $\boldsymbol{X} = [\boldsymbol{x}_1, \ldots, \boldsymbol{x}_n] \subseteq \mathbb{R}^d$ that is $(0, \beta_1)$-separable (with $\beta_1 < 0$). Furthermore, let the two-layer nonlinear DAE $\boldsymbol{f}_{\boldsymbol{W}_2, \boldsymbol{W}_1}(\boldsymbol{x})$ be overparameterized with $p \geq n$ hidden units. If we further assume the weight decay $\lambda$ in (6) satisfies $n\lambda < \min_{i \in [n]} \|\boldsymbol{x}_i\|_2^2$, then there exists a local minimizer of the DAE loss (6) with the following memorizing block-wise structure:*

$$\boldsymbol{W}_2^\star = \boldsymbol{W}_1^\star = (r_1 \boldsymbol{x}_1 \quad \cdots \quad r_n \boldsymbol{x}_n \quad \boldsymbol{0} \quad \cdots \quad \boldsymbol{0}) =: \boldsymbol{W}_{\text{mem}}, \quad r_i = \sqrt{\frac{\|\boldsymbol{x}_i\|_2^2 - n\lambda}{\|\boldsymbol{x}_i\|_2^4 + \sigma^2\|\boldsymbol{x}_i\|_2^2}}. \tag{9}$$

*Moreover, when $\lambda \to 0$, this solution attains an empirical loss that is independent of the ambient dimension $d$:*

$$\mathcal{L}_{\boldsymbol{X}}(\boldsymbol{W}_2^\star, \boldsymbol{W}_1^\star) = \frac{1}{n}\sum_{i=1}^n \frac{\sigma^2\|\boldsymbol{x}_i\|_2^2}{\sigma^2 + \|\boldsymbol{x}_i\|_2^2} < \sigma^2.$$

**Remarks.** The proof is deferred to Corollary D.3, and our result implies the following:

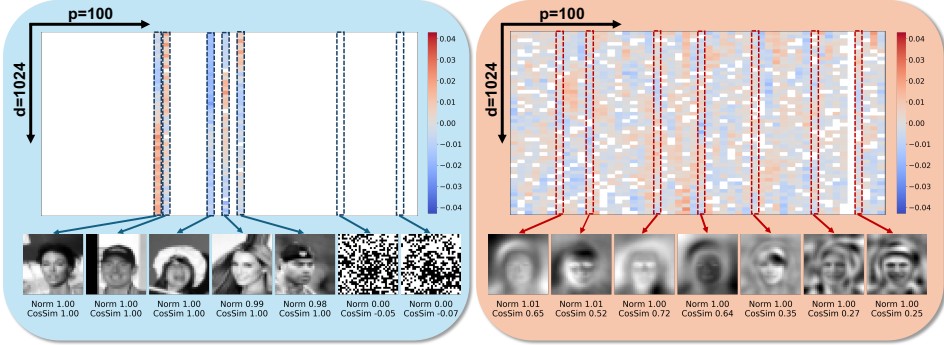

Figure 3: **Verification of Corollary 3.2 and Corollary 3.3.** We visualize the learned encoder matrix $\boldsymbol{W}_1$ of a ReLU DAE trained with noise level $\sigma = 0.2$. When trained on 5 CelebA face images, the model stores training samples in its columns, matching Corollary 3.2. When trained on 10,000 images, the model generalizes and captures data statistics, consistent with Corollary 3.3. Empirically, the same behavior holds for larger noise, up to $\sigma = 5$; additional results are in Appendix B.1.

- **Learning the optimal solution with sparse columns.** The structured solution with $(p - n)$ trailing zero columns in (9) is one among many local minimizers, as dense alternatives can arise by splitting sparse columns. However, empirical evidence and theory (Xie et al., 2025) suggest that standard optimizers such as Adam (Kingma and Ba, 2015) bias training toward $\ell_\infty$-smooth solutions of the DAE loss (cf. Corollary D.4). As a result, the solutions observed in practice often align with the sparse structure we construct (Figure 3).

- **Sampling reproduce training samples (memorization).** In this regime, the learned DAE closely approximates the empirical denoiser $\boldsymbol{f}_{\text{emp}}$ in Eq. (3), achieving low empirical loss and consequently reproducing the training samples under sampling (as shown in Figure 2). This occurs because the DAE's projection and reconstruction over the sparse columns of the weights during the reverse sampling effectively act as a power method, recovering memorized training data (Weitzner et al., 2024). Quantitatively, by plugging Corollary 3.2 into the overall denoising score matching loss, we find that the KL divergence between the sampled and empirical distributions is bounded by $\frac{\pi}{2} \max_{1 \le i \le n} \|\boldsymbol{x}_i\|$, confirming strong memorization.

- **Spiky representations as a signature of memorization.** As a consequence of Corollary 3.2, for any training sample $\boldsymbol{x}_i$, its learned representation within the DAE exhibits a distinctive sparse form:

$$\boldsymbol{h}_{\text{mem}}(\boldsymbol{x}_i + \sigma\boldsymbol{\epsilon}) = [\boldsymbol{W}_{\text{mem}}^\top(\boldsymbol{x}_i + \sigma\boldsymbol{\epsilon})]_+ \approx (0, \ldots, 0, \, r_i\boldsymbol{x}_i^\top(\boldsymbol{x}_i + \sigma\boldsymbol{\epsilon}), \, 0, \ldots, 0).$$

This sparsity arises because $\boldsymbol{x}_i$ is *negatively* correlated with other samples stored in the learned weight matrix $\boldsymbol{W}_{\text{mem}}$, yielding a nearly one-hot feature vector within the representation space (Figure 4). Such *spikiness* could serve as a robust signature of memorization (Hakemi et al., 2025; Gan et al., 2025), which we empirically demonstrate on both synthetic (Figure 4) and real-world (Figure 5) settings. Building on this insight, we introduce a simple yet effective memorization detection method that achieves strong results, as detailed later in Section 4.1. Additionally, analogous correlations between sharp, localized activations and the recall of concrete stored knowledge have been empirically observed in Large Language Models (LLMs) (Sun et al., 2024), suggesting our findings could also offer a potential explanation for these phenomena in LLMs.

## 3.2 CASE 2: GENERALIZATION WITH UNDERPARAMETERIZATION

On the other hand, suppose we have sufficiently many i.i.d. samples $\{\boldsymbol{x}_{k,i}\}_{i=1}^{n_k}$ from each Gaussian mode $k \in [K]$ of the MoG distribution (4). Then the empirical mean and Gram matrix of each cluster $k$ concentrate around their expectations:

$$\overline{\boldsymbol{x}}_k = \frac{1}{n_k}\sum_{i=1}^{n_k}\boldsymbol{x}_{k,i} \approx \boldsymbol{\mu}_k, \quad \frac{1}{n_k}\boldsymbol{X}_k\boldsymbol{X}_k^\top \approx \boldsymbol{S}_k := \boldsymbol{\mu}_k\boldsymbol{\mu}_k^\top + \boldsymbol{\Sigma}_k. \tag{10}$$

If the component means are incoherent (i.e., $\langle\boldsymbol{\mu}_k, \boldsymbol{\mu}_\ell\rangle/(\|\boldsymbol{\mu}_k\|\|\boldsymbol{\mu}_\ell\|) < \beta_2$ for $k \ne \ell$) and the within-mode variance is small (i.e., $\|\boldsymbol{\Sigma}_k^{1/2}\|_F/\|\boldsymbol{\mu}_k\|_2 < \alpha_2$), then with high probability the clusters $\{\boldsymbol{X}_k\}_{k=1}^K$ satisfy the separability conditions in Definition 3.1 with $(\alpha, \beta) = (\alpha_2, \beta_2)$. In this scenario, as we demonstrate below, the optimal weights of the DAE network will learn the local

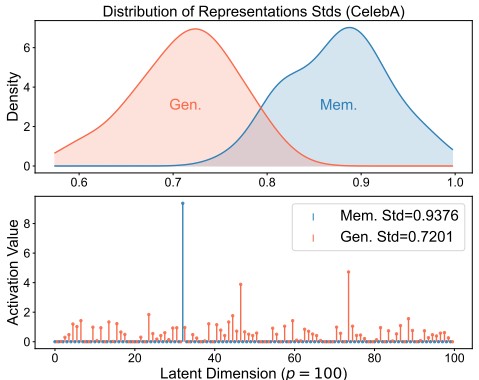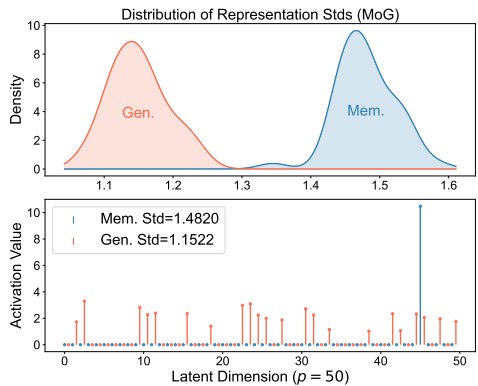

Figure 4: **Mem./Gen. representations in ReLU DAEs.** *Top:* Memorized vs. generalized samples can be separated by the standard deviation (Std) of their representations: memorized models produce spiky, high-Std features, whereas generalized models do not. *Bottom:* Representation of a single training data. The memorized model exhibits large outlier activations (high Std); the generalized model yields a more balanced representation (lower Std), consistent with our theory. All models use $\sigma = 0.2$. Left: CelebA. Right: MoG. See Appendix C.1 for details.

data statistics (specifically, the means and variances of the MoG) from these well-separated, non-degenerate clusters of training data to enable generalization.

**Corollary 3.3** (Generalization in Underparameterized DAEs). *Under the problem setup of Theorem 3.1, we assume the training data satisfy the separability condition in Definition 3.1.[1] If the DAE network in* (5) *is under-parameterized with $p = \sum_{k=1}^{K} p_k \ll n$, then there exists a local minimizer of the DAE training loss* (6) *such that*

$$\boldsymbol{W}_2^{\star} = \boldsymbol{W}_1^{\star} = (\boldsymbol{W}_{\boldsymbol{X}_1} \quad \boldsymbol{W}_{\boldsymbol{X}_2} \quad \cdots \quad \boldsymbol{W}_{\boldsymbol{X}_K}) =: \boldsymbol{W}_{\text{gen}},$$

*where each block $\boldsymbol{W}_{\boldsymbol{X}_k} \in \mathbb{R}^{d \times p_k}$ captures the principal components of the empirical Gram matrix $\boldsymbol{X}_k \boldsymbol{X}_k^{\top}$ in* (8)*, with $\boldsymbol{W}_{\boldsymbol{X}_k} \boldsymbol{W}_{\boldsymbol{X}_k}^{\top}$ concentrating to the rank-$p_k$ optimal denoiser for $\mathcal{N}(\boldsymbol{\mu}_k, \boldsymbol{\Sigma}_k)$:*

$$\boldsymbol{W}_{\boldsymbol{X}_k} \boldsymbol{W}_{\boldsymbol{X}_k}^{\top} \to \left[ \left(\boldsymbol{S}_k - \frac{\lambda}{\rho_k}\boldsymbol{I}\right)(\boldsymbol{S}_k + \sigma\boldsymbol{I})^{-1} \right]_{\text{rank-}p_k \text{ approx}},$$

*where $\boldsymbol{S}_k$ is introduced in* (10) *and $\rho_k$ is the ratio of the $k$-th mode of MoG. Moreover, when $\lambda \to 0$, the expectation of the test loss (which captures generalization error) can be bounded by*

$$\mathbb{E}_{\boldsymbol{X} \sim p_{gt}}[\mathcal{L}_{\boldsymbol{X}}(\boldsymbol{W}_2^{\star}, \boldsymbol{W}_1^{\star})] \lesssim \sum_{k=1}^{K} \rho_k \left[ \sum_{j \leq p_k} \frac{\text{eig}_j(\boldsymbol{S}_k) \cdot \sigma^4}{\left(\text{eig}_j(\boldsymbol{S}_k) + \sigma^2\right)^2} + \sum_{j > p_k} \text{eig}_j(\boldsymbol{S}_k) + \frac{C_k \, p_k}{\sigma^2 \, n_k} \right],$$

*where $C_k > 0$ depends only on $\sigma$ and spectral properties of $\boldsymbol{S}_k$. Here, $\text{eig}_j(\boldsymbol{S}_k)$ denotes the $j$-th eigenvalue of $\boldsymbol{S}_k$ which is independent of $d$.*

**Remarks.** The proof is deferred to Appendix D.5, and our result implies the following:

- **Sampling yields novel in-distribution samples (generalization).** When the model is under-parameterized, our results show that the local optimal solution learned from the training data achieves bounded population loss on the MoG distribution by effectively acting as an optimal local denoiser for each mode. Consequently, sampling (Li et al., 2024a; 2025a) from the trained DAE produces in-distribution images that are distinct from the training samples, as illustrated in Figure 2.

  Moreover, the population loss depends on the spectrum of $\boldsymbol{S}_k$ (equivalently, $\boldsymbol{\Sigma}_k$). When $\boldsymbol{\Sigma}_k$ has an approximately low-rank structure (De Bortoli, 2022; Cole and Lu, 2024; Huang et al., 2024), the loss is small and decays rapidly with the number of samples per mode $n_k$. This provides a principled explanation for the reproducibility of diffusion models across disjoint training subsets (Zhang et al., 2024; Kadkhodaie et al., 2024a).

- **Balanced representations as a signature of generalization.** Unlike the spiky representations in Corollary 3.2, the underparameterized solution spreads the energy of $\boldsymbol{x}_i + \sigma\boldsymbol{\epsilon}$, with $\boldsymbol{x}_i \sim$

---

[1]For simplicity, we take separability as an assumption; given sufficient samples, it can be verified under extra conditions on the means and covariances of MoG using standard measure concentration (Vershynin, 2018).

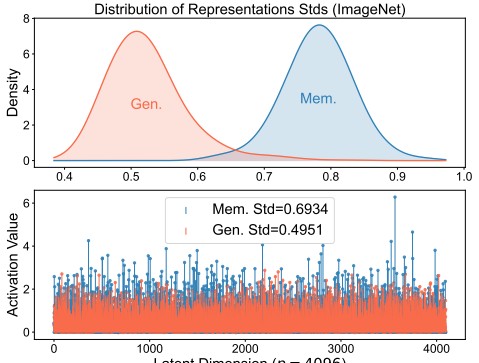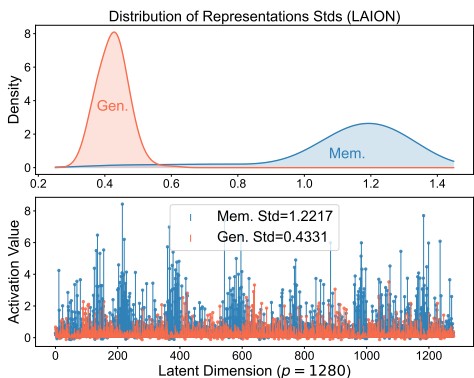

Figure 5: **Mem./Gen. representations in real-world models.** memorized samples have spiky representations while generalized samples have more balanced ones. The layout follows Figure 4 and the results are consistent with it. Representations are extracted at timestep $t = 50$ ($\sigma_t \approx 0.17$). *Left*: DiT-L/4 pretrained on an ImageNet subset. *Right*: Stable Diffusion v1.4 pretrained on LAION (Schuhmann et al., 2022). Results for EDM pretrained on CIFAR10 and additional details are in Appendix C.2.

$\mathcal{N}(\boldsymbol{\mu}_k, \boldsymbol{\Sigma}_k)$, across the $p_k$ coordinates of the active block (see Figure 4). The representation behaves like a **low-dimensional projection for a Gaussian mode (Tipping and Bishop, 1999)**:

$$\boldsymbol{h}_{\text{gen}}(\boldsymbol{x}_i + \sigma\boldsymbol{\epsilon}) = [\boldsymbol{W}_{\text{gen}}^\top(\boldsymbol{x}_i + \sigma\boldsymbol{\epsilon})]_+ \approx (0, \ldots, 0, \boldsymbol{W}_{\boldsymbol{X}_k, 1}^\top(\boldsymbol{x}_i + \sigma\boldsymbol{\epsilon}), \ldots, \boldsymbol{W}_{\boldsymbol{X}_k, p_k}^\top(\boldsymbol{x}_i + \sigma\boldsymbol{\epsilon}), 0, \ldots, 0).$$

Intuitively, generalized samples activate multiple neurons rather than a single spiky unit; the resulting projections encode information about the underlying distribution, helping to explain empirical findings on semantic directions (Kwon et al., 2023) that are useful for editing, which we further explore in Section 4.2.

Concluding Corollary 3.2 and Corollary 3.3, we see the learned structure remains stable across timesteps, with $\sigma$ primarily acting as a regularization parameter. Varying $\sigma$ only slightly perturbs the solution, which helps explain the empirical success of diffusion models that employ a single neural network for denoising across multiple noise levels (Sun et al., 2025).

### 3.3 CASE 3: HYBRID OF MEMORIZATION AND GENERALIZATION WITH IMBALANCED DATA

Large-scale diffusion datasets often contain duplicates due to imperfect curation or heterogeneous aggregation (Carlini et al., 2023); such samples are more easily memorized (Somepalli et al., 2023b) (See Appendix B.4 for more discussions). We model this by allowing duplicated (rank-1) clusters alongside well-sampled, nondegenerate clusters, so the DAE can admit local minimizers that mix memorization and generalization blocks:

**Corollary 3.4** (DAE memorizes duplicates and generalizes on well-sampled modes). *Let $X = [\boldsymbol{X}_1, \ldots, \boldsymbol{X}_K]$ satisfy Definition 3.1, where for $\ell = 1, \ldots, m$, $\boldsymbol{X}_\ell = (\boldsymbol{x}_\ell, \ldots, \boldsymbol{x}_\ell)$ is rank 1, and $\boldsymbol{X}_{m+1}, \ldots, \boldsymbol{X}_K$ contain distinct empirical samples from the remaining Gaussian modes. Suppose a ReLU DAE is trained with weight decay $\lambda \geq 0$ and input noise $\sigma > 0$. Then there exists a local minimizer of the form*

$$\boldsymbol{W}_2^\star = \boldsymbol{W}_1^\star = (r_1\boldsymbol{x}_1 \quad \cdots \quad r_m\boldsymbol{x}_m \quad \boldsymbol{W}_{\boldsymbol{X}_{m+1}} \quad \cdots \quad \boldsymbol{W}_{\boldsymbol{X}_K}),$$

*where the first $m$ columns memorize the duplicated clusters (as in Cor. 3.2), and the remaining blocks $\boldsymbol{W}_{\boldsymbol{X}_k}$ implement generalization on the nondegenerate clusters (as in Cor. 3.3).*

This corollary interpolates Cases 1 and 2: duplicated training samples are memorized, while the model still generalized for the other modes. We verify this in Figure 6 and defer proof to Appendix D.6.

## 4 IMPLICATIONS FOR MEMORIZATION DETECTION AND CONTENT STEERING

In this section, we demonstrate that our theoretical insights from Section 3 yield profound practical implications for model privacy and interpretability. Leveraging the identified dual relationship between representation structures and generalization ability, we present the following two applications:

Figure 6: **Verification of Corollary 3.4**. The model learns both memorizing and generalizing columns when data duplication is present.

| Method | Prompt Free? | LAION | | | ImageNet | | | CIFAR10 | | |
|---|---|---|---|---|---|---|---|---|---|---|
| | | AUC ↑ | TPR ↑ | Time ↓ | AUC ↑ | TPR ↑ | Time ↓ | AUC ↑ | TPR ↑ | Time ↓ |
| (Carlini et al., 2023) | ✗ | 0.498 | 0.020 | 3.724 | | N/A | | | N/A | |
| (Wen et al., 2024) | ✗ | 0.986 | **0.961** | 0.134 | | N/A | | | N/A | |
| (Hintersdorf et al., 2024) | ✗ | 0.957 | 0.500 | **0.009** | | N/A | | | N/A | |
| (Ross et al., 2025) | ✓ | 0.956 | 0.915 | 0.545 | 0.971 | 0.528 | 0.031 | 0.713 | 0.013 | 0.071 |
| Ours | ✓ | **0.987** | **0.961** | 0.067 | **0.995** | **0.912** | **0.015** | **0.998** | **0.984** | **0.020** |

Table 1: **Memorization detection results.** We report AUROC, true positive rate (TPR) at 1% false positive rate, and runtime (s). Evaluated on three dataset-model pairs: LAION-SD1.4, ImageNet-DiT, and CIFAR10-EDM. Sample sizes: 500 memorized and 500 generalized for LAION and ImageNet; 100 each for CIFAR10. (↑ higher is better; ↓ lower is better). See Appendix C.2 for details.

- **Representation-based memorization detection (Section 4.1).** Leveraging the spikiness of data representations, we introduce a prompt-free classification method that accurately distinguishes between generalized and memorized samples produced by diffusion models. We demonstrate that our approach achieves strong performance with high efficiency and extensibility.

- **Representation-space steering for image editing (Section 4.2).** We introduce a training-free editing method that steers generated samples within the representation space. Crucially, we find that generalized samples are substantially more steerable, whereas memorized samples exhibit minimal editing effects due to the spikiness of their representations.

## 4.1 REPRESENTATION-BASED MEMORIZATION DETECTION

Building on our theoretical insights, we investigate whether memorization can be **detected** directly from internal representations. Prior work has largely focused on how certain prompts trigger memorization and often relies on those for detection (Wen et al., 2024; Jeon et al., 2025; Ren et al., 2024). Representative approaches include: (*i*) *Density-based:* detecting samples that are generated disproportionately frequently under a prompt (Carlini et al., 2023); and (*ii*) *Norm-based:* comparing conditional vs. unconditional scores (Wen et al., 2024) and (*iii*) *Attention-based:* locating anomaly in the cross-attention induced by memorized prompts (Hintersdorf et al., 2024; Chen et al., 2025a). A notable exception is (*iv*) a *landscape-based* method of (Ross et al., 2025), which evaluates memorization using local score-function geometry around a generated sample. Their method makes detection prompt-free, but is still based on output space.

In contrast, we introduce the first detection method that is both **representation-based** and **prompt-free**. The core intuition is that *spiky representations* arise when a sample has been internally stored by the model, whereas generalized samples yield balanced activations. Therefore, our analysis yields a simple yet effective diagnostic: the standard deviation of intermediate features serves as a proxy for spikiness. High variance indicates memorization; low variance corresponds to generalization. We benchmark this detector against existing baselines on pre-trained diffusion models. As reported in Table 1, our method achieves the highest accuracy and efficiency, thereby demonstrating the strong informativeness of representation-space statistics. Pseudocode and further implementation details are provided in Appendix C.2.

## 4.2 REPRESENTATION-SPACE STEERING FOR INTERPRETABLE IMAGE EDITING

As shown in Corollary 3.3, representations of generalized samples are governed by data statistics, capturing local semantics and acting as low-dimensional projections of Gaussian modes. This insight implies an interpretable steering mechanism: we can inject information about a target mode (e.g., a specific concept or style) by adding its average representation, thereby smoothly guiding

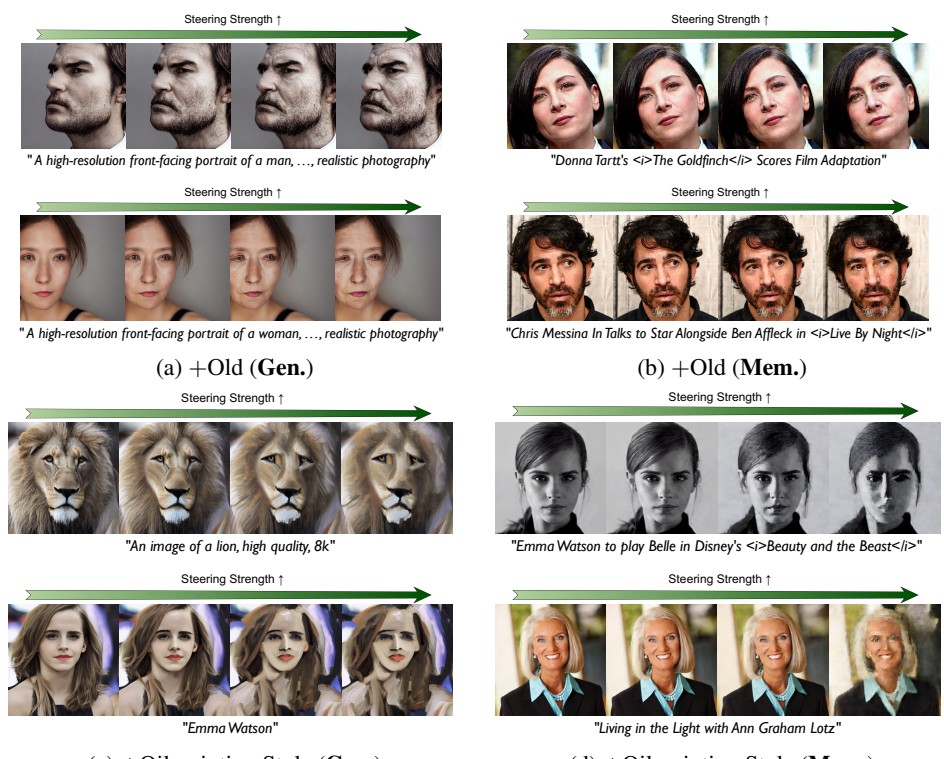

Figure 7: **Image editing via representation steering.** We perform image editing on Stable Diffusion v1.4 using (11). Generalized samples exhibit smooth and progressive style transfer as the editing strength increases, whereas memorized samples display brittle and threshold-like transfer effects.

generation toward it. Specifically, our proposed steering function is defined as:

$$\boldsymbol{f}_{\boldsymbol{\theta}}^{\text{steered}}(\boldsymbol{x}_t, t, c) = \boldsymbol{g}_{\boldsymbol{\theta}}(\boldsymbol{h}_{\boldsymbol{\theta}}(\boldsymbol{x}_t, t, c) + a\boldsymbol{v}), \quad \text{where} \quad \boldsymbol{v} = \frac{1}{|\mathcal{S}|} \sum_{\tilde{\boldsymbol{x}} \in \mathcal{S}} \boldsymbol{h}_{\boldsymbol{\theta}}(\tilde{\boldsymbol{x}}_{\tilde{t}}, \tilde{t}, \bar{c}). \quad (11)$$

Here, $\mathcal{S}$ denotes samples from the target concept/style, $\boldsymbol{h}_{\boldsymbol{\theta}}$ and $\boldsymbol{g}_{\boldsymbol{\theta}}$ represent the encoder and decoder components of the network, respectively, and $a \in \mathbb{R}$ controls the steering strength, $c$ is the text prompt and $\bar{c}$ denotes the desired concept/style prompt.

We evaluate this method on Stable Diffusion v1.4 using both memorized and generalized samples (Figure 7). As predicted by our theory, generalized samples exhibit smooth and monotonic edits as $a$ varies, indicative of a well-behaved local geometry in their representation space. In contrast, memorized samples display brittle, threshold-like responses, making fine-grained control difficult because of their spiky representations.

## 5 CONCLUSION

In summary, our study establishes that the representation space of diffusion models is not a secondary artifact of training but a critical factor in how these models operate. Its structure provides a principled separation between memorization and generalization: spiky, sample-specific codes signal memorization, while balanced, low-dimensional representations often imply strong generalization. This perspective allows us not only to detect memorization directly from internal model representations but also to leverage representations for practical tasks, such as controllable editing via steering. While prior works have used intermediate activations for downstream applications, our framework highlights their important role in shaping diffusion behavior itself. By making these structures explicit, we bridge the theoretical findings on simplified models with the empirical properties of real-world deep nonlinear models, offering a unified view that connects perception and generation and opens pathways toward more interpretable and trustworthy generative models.

## ACKNOWLEDGMENTS

We acknowledge funding support from NSF CAREER CCF-2143904, NSF CCF-2212066, NSF CCF-2212326, NSF IIS 2402950, ONR N000142512339, DARPA HR0011578254, and Google Research Scholar Award. We also thank Shuo Xie (TTIC; for fruitful discussions on optimization bias), SooMin Kwon, Huijie Zhang, Alec Xu, Prof. Laura Balzano (UMich; for help with problem formulation), Prof. Zhihui Zhu (OSU; for guidance on presentations), Zaicun Li (UMD; for mathematical insights), and Zhenyun Zhu (UMich; for scientific discussions).

## CODES OF ETHIC AND REPRODUCIBILITY

**Ethics Statement.** This study utilizes only theoretical frameworks and controlled experiments on publicly available datasets and research models; no human subjects, sensitive data, or personally identifiable information are involved. We acknowledge dual-use risks: memorization detection could be misused to extract or target memorized content, and representation steering could be abused to generate harmful outputs. Our intent is the opposite: to improve auditability and safety by (i) detecting and characterizing memorization, and (ii) observing (and then disabling) harmful steering directions at the post-training stage. All experiments were conducted under controlled conditions using standard public memorization benchmarks, adhering to dataset licenses and established practices as in previous memorization-detection literature. Code and artifacts, if released, will include safeguards and usage guidelines to discourage misuse.

**Reproducibility Statement.** All assumptions underlying our theoretical results are explicitly stated in the main paper. Proofs are provided in Appendix D, with additional verification in Appendix B.1. Empirical details are provided in Appendix C, and code is included with the submission to support reproducibility.

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

## A    ADDITIONAL RELATED WORKS

### A.1    ANALYSIS OF LEARNING DIFFUSION MODELS WITH SPECIFIC MODEL PARAMETERIZATIONS

There has been a large body of work analyzing the learning of diffusion models (Wang et al., 2024b; Chen et al., 2023; Liang et al., 2025; Yang et al., 2024). Recently, more attention has turned to when and how they overfit or generalize: Li et al. (2023); Bonnaire et al. (2025) use random-feature assumptions, while Wang et al. (2024a); Buchanan et al. (2025) studied empirical denoisers with learnable attractors; Wu et al. (2025); Chen (2025); Ye et al. (2025) investigated smoothing effects induced by learning rates and weight decay that promote generalization. These works are theoretically rigorous but often lack real-world validation.

### A.2    MEMORIZATION AND GENERALIZATION WITH ANALYTICAL DMS

**Constrained/regularized models.**    Recent works characterize how architectural or inductive biases can push empirical scores toward more generalizable solutions. For instance, Scarvelis et al. (2023); Lukoianov et al. (2025) constructed closed-form diffusion models from data; Niedoba et al. (2024; 2025); Kamb and Ganguli (2025); Wang et al. (2025b) imposed locality or translation-equivariance constraints to mimic U-Net behavior; and Kadkhodaie et al. (2024a); An et al. (2025); Floros et al. (2025) analyzed architectural biases of CNNs and DiTs. Baptista et al. (2025) empirically evaluated the impact of various regularization schemes.

**Associative Memory (AM) models.**    Radhakrishnan et al. (2020); Ambrogioni (2024); Pham et al. (2025) model imperfect training and sampling jointly as an AM recall process, viewing novel image generation as new attraction basins and memorization as perfect recalls (Biroli et al., 2024; Lyu et al., 2025). However, this perspective can understate the role of learned neural networks in enabling generalization.

### A.3    STUDIES ON REPRESENTATION LEARNING OF DIFFUSION MODELS

Concurrent work studies co-emerging representation learning (Kwon et al., 2023; Han et al., 2025; Yang and Wang, 2023) with distribution learning in diffusion models. As recent works (Chen et al., 2025b; Xiang et al., 2025) re-emphasize that the diffusion objective is fundamentally a self-supervised autoencoder loss (Vincent et al., 2010; Vincent, 2011; Bengio et al., 2013), which induces encoder-decoder behavior (Chen et al., 2025b) and the model autonomously learns informative features for downstream tasks (Baranchuk et al., 2022; Xiang et al., 2023; Zhao et al., 2023). Moreover, supervising the representations can accelerate training (Yu et al., 2025; Wang et al., 2025a; Singh et al., 2025), and different representation behaviors correlate with different degrees of overfitting (Li et al., 2025c).

## B    ADDITIONAL EXPERIMENTS

### B.1    FURTHER VERIFICATION OF THEOREM 3.1

**Verification with MoG data**    Under a Mixture of Gaussians (MoG) setting, we directly verify Theorem 3.1 since the separability assumptions can be enforced by construction. We use a two-mode MoG in a 1000-dimensional space with symmetric means $\boldsymbol{\mu}_1 = -\boldsymbol{\mu}_2 = 5\boldsymbol{e}_1$, where $\boldsymbol{e}_1 = (1, 0, \ldots, 0)$, and covariance matrices $\boldsymbol{\Sigma}_1, \boldsymbol{\Sigma}_2$ each having exponentially decaying spectra. We sample 5,000 points from each mode, yielding two separated clusters. Training a ReLU DAE with $p = 50$ hidden units and $\sigma = 0.2$, we find that the model effectively learns a rank-25 approximation of the Wiener filter for each cluster, as defined in Theorem 3.1, as shown in Figure 8:

**Robustness to large noise levels**    We show here that the vanishing remainder in Theorem 3.1 is negligible even for large $\sigma$s. For instance, we train the ReLU DAE under $\sigma = 0.2, 1, 5$ on CelebA and we find the model still learns the constructed solution as shown in Figure 9.

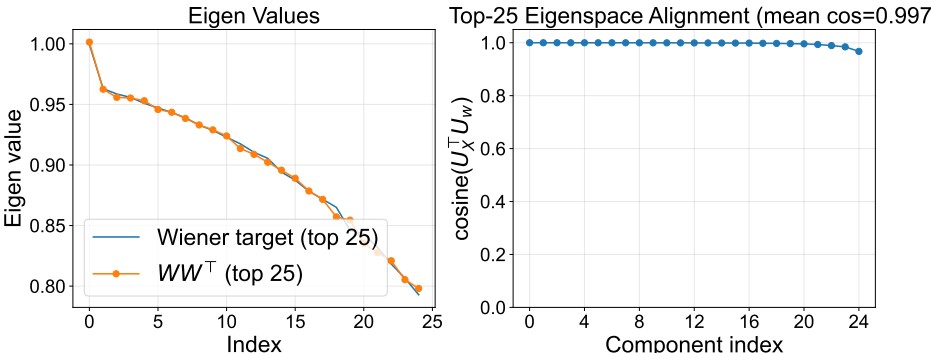

Figure 8: Comparison between the learned ReLU DAE and the constructed solution from Theorem 3.1 under the MoG setting. They agree in both eigenvalues and eigenvectors.

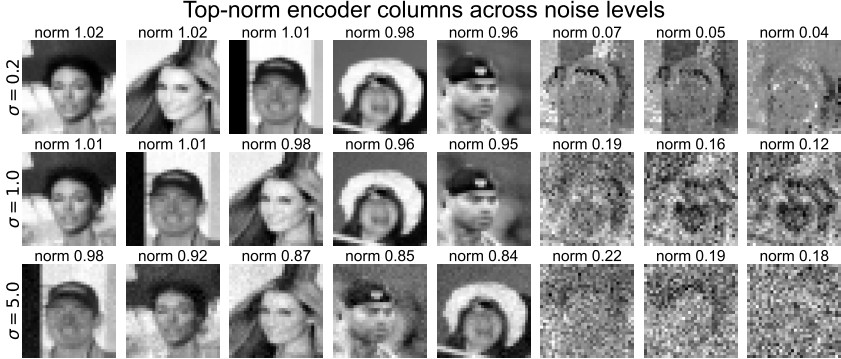

Figure 9: training with larger $\sigma$ will give us less perfect memorization, but the trend holds

**Robustness beyond separability**   When the separability assumption is relaxed ($\beta > 0$, so training images overlap), the memorized ReLU DAE still learns a processed version of the solution in Theorem 3.1. Empirically, it recovers a denoised/processed data matrix, or approximately an orthonormal basis for the data span; see Figure 10.

For the generalized ReLU DAE on CelebA, it is already a non-separable dataset. And the model continues to (i) generate novel images (Figure 2), (ii) capture dataset statistics (Figure 3), and (iii) produce balanced representations (Figure 4). Thus, separability mainly simplifies the form of local minimizers; it is not required for either memorization or generalization.

**Robustness to different optimization setups**   We show that the local minimizer characterized in Cor. 3.2 is robust to different random seeds and optimizers (RMSProp, Adam, AdamW). In all cases, modern adaptive optimizers converge to a sparse solution that stores individual training samples as columns. We also varied the random seed and found that it essentially only permutes the columns, and omit those results for brevity (Figure 11).

**Tying vs. untying the encoder-decoder matrices**   Our theorem shows that even when the encoder and decoder are parametrized independently, training drives them to a symmetric (tied) solution. We confirm this empirically in Figure 12, consistent with prior observations (Kunin et al., 2019). Accordingly, for Figures 3 and 4 in the main text we train weight-tied ReLU DAEs.

### B.2   DENOISING AND REPRESENTATIONS OF TEST SAMPLES WITH RELU DAE

As in (Kadkhodaie et al., 2024a)), the ability to denoise an unseen test image is an equivalent check for generalization or overfitting, as shown in Figure 13. Memorizing DAE (Corollary 3.3) perfectly denoises a training image. On a test image, it still produces a training-data like output (visu-

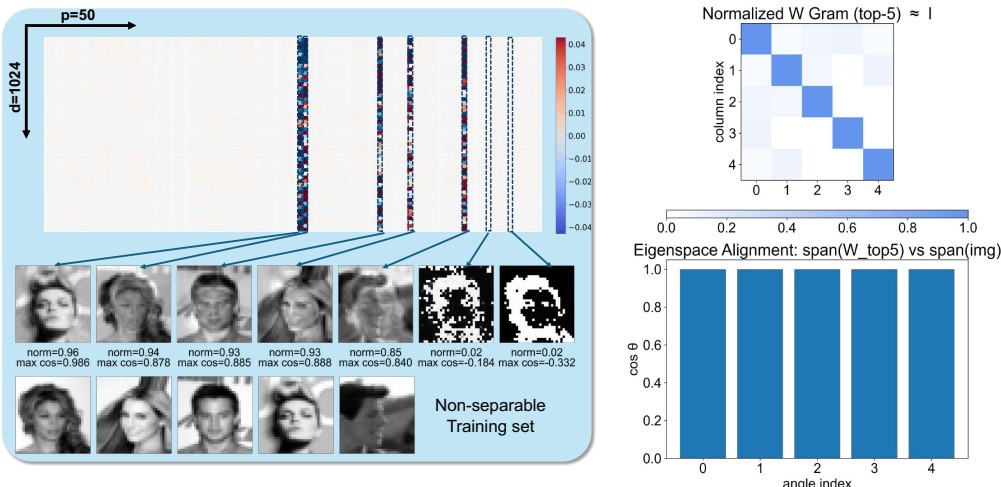

Figure 10: When separability breaks, a ReLU DAE still learns a processed version of the data matrix (approximately an orthonormal basis of the data span).

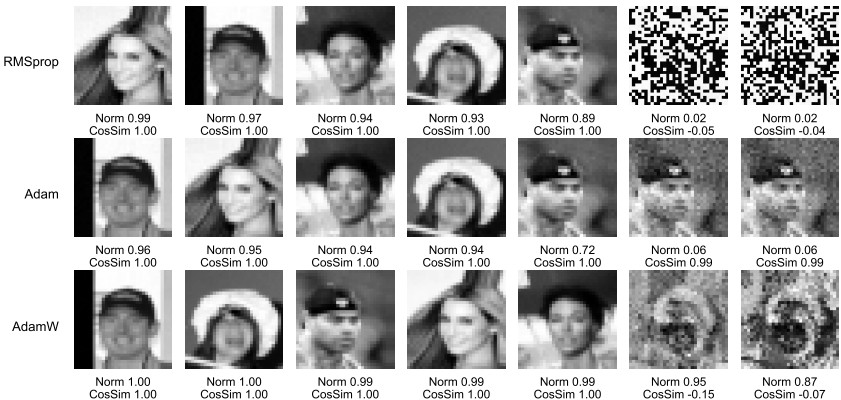

Figure 11: The local minimizer from Cor. 3.2 is robust to different random seeds and optimizers.

ally clear but discarding input-specific information, producing high test MSE). Generalizing DAE (Corollary 3.3) denoises both while preserving input-specific structure.

Moreover, we also visualize the representations of test samples for the memorizing and generalizing DAEs in Figure 14. Since the memorizing DAE learns sparse columns, the representation of a test image is also sparse: positive activations indicate positive alignment with specific memorized training samples, and the resulting code is highly spiky. For the generalizing DAE, which learns statistics reflecting the underlying data distribution, the representations of test samples are as balanced as those of training samples.

### B.3 CONNECTION BETWEEN RELU DAE AND REAL-WORLD MODELS

In this section, we demonstrate that our ReLU model exhibits piecewise linearity, consistent with observations in real-world models (Lukoianov et al., 2025). Consequently, it can be viewed as a localized approximation of these counterparts: a large model can implement the mechanisms of Corollary 3.2 and Corollary 3.3 in distinct local regions (Ross et al., 2025), thereby simultaneously generalizing and memorizing. We verify this via SVD analysis of the Jacobian (Kadkhodaie et al., 2024a; Achilli et al., 2024) for SD1.4, EDM, and our ReLU DAE:

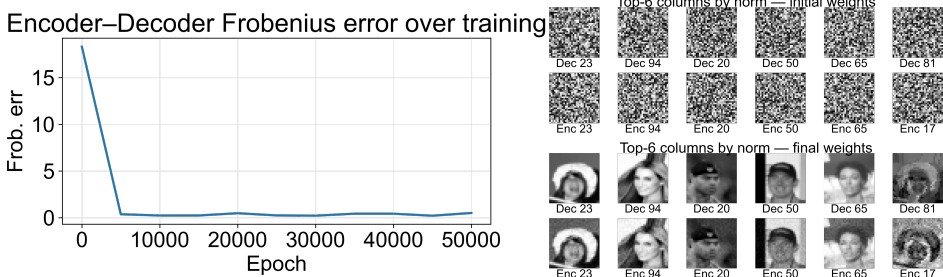

Figure 12: An untied ReLU DAE learns (approximately) symmetric encoder-decoder matrices.

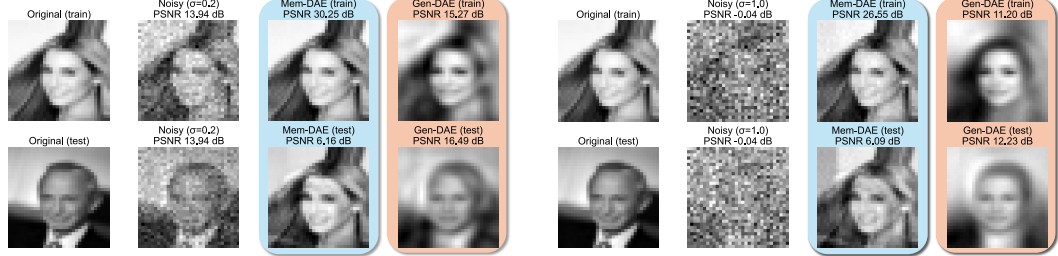

(a) Denoising $\sigma = 0.2$ with Mem./Gen. DAEs   (b) Denoising $\sigma = 1.0$ with Mem./Gen. DAEs

Figure 13: One-step denoising result of train/test samples with ReLU DAE

- Around memorized data, the model's Jacobian is extremely low-rank and dominated by **that** specific data vector. This indicates the model is storing and denoising along the memorized sample, confirming the results of Cor. 3.2. Moreover, the model denoises with near-perfect certainty.

- Around generalized samples, the Jacobian matrix reflects the data structures described in corollary 3.3. Accordingly, the model produces a smoothed result, having learned a ground-truth denoiser that incorporates the constraints of the underlying distribution (Niedoba et al., 2025).

We visualize these findings in Figures 15a, 15b, and 15c.

### B.4 DUPLICATION OF TRAINING DATA INDUCES MEMORIZATION

Large-scale diffusion datasets often contain duplicates due to imperfect deduplication or aggregation from heterogeneous sources (Carlini et al., 2023; Shi et al., 2025). Such duplicates are disproportionately memorized by generative models (Somepalli et al., 2023b; Chen et al., 2024a). Interpolating Corollary 3.2 and Corollary 3.3 suggests that, when a subset is duplicated, the model tends to memorize those duplicated samples while still generalizing on the rest. We observe this behavior empirically in Figure 16 for EDM trained on CIFAR10 with a duplicated subset (and similarly for DiT on ImageNet as in Figure 5 ).

## C EXTRA TECHNICAL DETAILS

### C.1 TRAINING AND SAMPLING SETUP FOR ReLU DAEs

**Optimization.** We train with RMSprop. For memorized models, we use learning rate $1 \times 10^{-3}$, weight decay $1 \times 10^{-2}$, and run $5 \times 10^5$ gradient steps. For generalized models, we use learning rate $1 \times 10^{-4}$, weight decay $1 \times 10^{-4}$, and run $4 \times 10^7$ steps. Perturbing these choices (e.g., Adam/AdamW vs. RMSprop, slightly different learning rates or weight decays, or tying vs. untying the encoder-decoder) can slightly shift the final solution, but the memorization-generalization characterization remains clear.

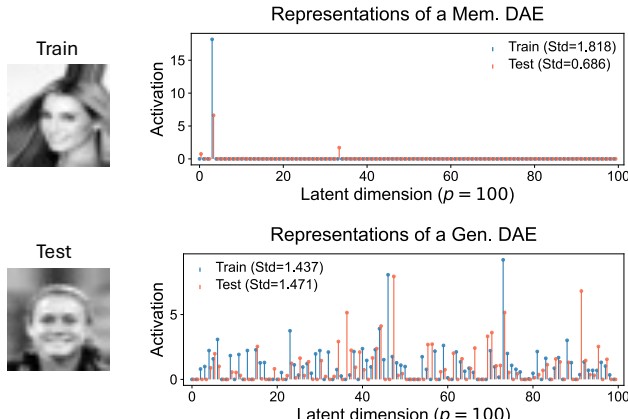

Figure 14: Representations of train and test samples under memorizing vs. generalizing ReLU DAEs.

**Sampling.** We train a set of DAEs with VE noise scheduling (Song et al., 2021b) over $\sigma \in [0.02, 2]$ and run DDIM sampling (Eq. 1).

**Data.** *MoG:* In $d = 1000$, we consider two symmetric modes with means $\boldsymbol{\mu}_1 = -\boldsymbol{\mu}_2 = 5\mathbf{e}_1$ (where $\mathbf{e}_1 = (1, 0, \ldots, 0)$) and covariances $\boldsymbol{\Sigma}_1, \boldsymbol{\Sigma}_2$ having exponentially decaying spectra. For the memorized model, we use 2 samples per mode; for the generalized model we use 10,000 samples (5,000 per mode). *CelebA:* We use 5 training images (chosen for clear separability) for the memorized model and the first 10,000 for the generalized model.

## C.2 MEMORIZATION DETECTION DETAILS

**Collecting mem./gen. sets.** For LAION-Stable Diffusion, we follow (Wen et al., 2024) and use publicly available prompts curated to elicit either memorization or generalization (Webster et al., 2023). For CIFAR10-EDM and ImageNet-DiT, we compute the SSCD similarity (Zhang et al., 2024) between each generated image and its nearest neighbor in the training set; samples with similarity $> 0.9$ are labeled memorized and those with similarity $< 0.5$ as generalized.

**Feature extraction.** For EDM we extract activations at `8x8_block3.norm0`; for Stable Diffusion v1.4 at `up_blocks.0.resnets.2.nonlinearity`; and for DiT-L/4 we use the SiLU activation in block 12 (of 24). We apply global max pooling (spatial for Stable Diffusion v1.4 and EDM; token-wise for DiT) to obtain compact representations, though detection also works even if not. Unless otherwise noted, representations are taken at DDPM timestep $t = 50$, corresponding to an equivalent noise level $\sigma_t \approx 0.17$ (Ho et al., 2020).

---

**Algorithm 1:** Detection via representation standard deviation (STD)

---

**Input:** generated image $\boldsymbol{x}_0$, timestep $t$, threshold THRES
**Output:** intermediate representation $\boldsymbol{h}$, detection flag $\mathbb{I}_{\text{mem}}$
$\boldsymbol{x}_t \leftarrow \text{ADDFORWARDNOISE}(\boldsymbol{x}_0, t)$;
$\boldsymbol{h} \leftarrow \boldsymbol{h}_\theta(\boldsymbol{x}_t, t, \text{condition} = \varnothing)$;
 where $\boldsymbol{f}_\theta(\boldsymbol{x}_t, t) = \boldsymbol{g}_\theta[\boldsymbol{h}_\theta(\boldsymbol{x}_t, t, \varnothing)]$ with $\boldsymbol{g}$ and $\boldsymbol{h}$ the decoder/encoder components;
$\mathbb{I}_{\text{mem}} \leftarrow (\text{STD}(\boldsymbol{h}) > \text{THRES})$;
**return** $\boldsymbol{h}, \mathbb{I}_{\text{mem}}$;

---

The detection metric need not be limited to standard deviation; other effective choices include the $\ell_4/\ell_2$ ratio (Vershynin, 2018), entropy, and max-min of the representations. We found these alternatives yield similar separability between memorized and generalized samples.

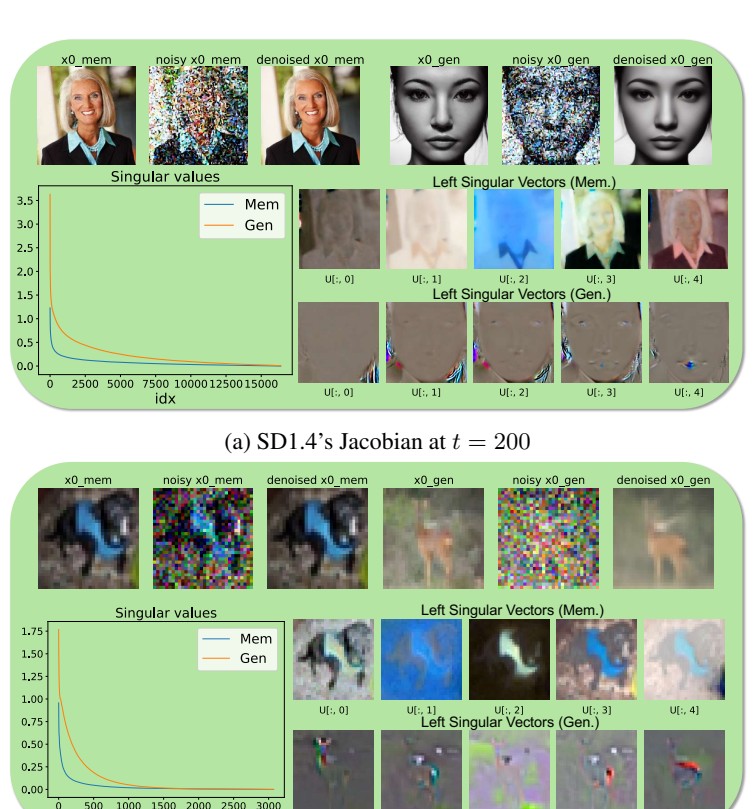

(a) SD1.4's Jacobian at $t = 200$

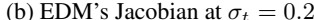

(b) EDM's Jacobian at $\sigma_t = 0.2$

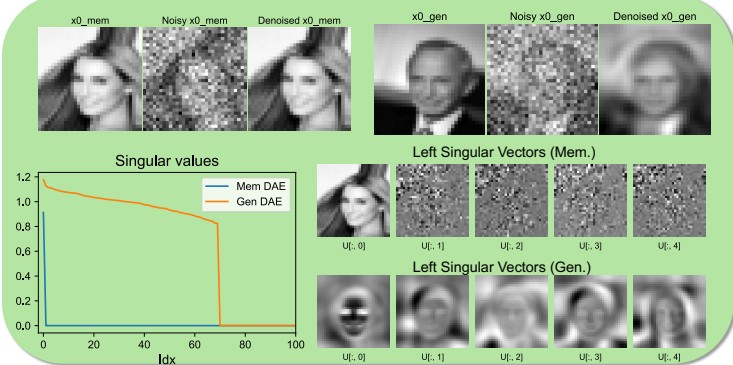

(c) ReLU DAE's Jacobian at $\sigma_t = 0.2$

Figure 15: Jacobians for SD1.4, EDM, and ReLU DAE at the indicated time/noise settings.

### C.3 IMAGE EDITING DETAILS

We use Stable Diffusion v1.4 for our image editing experiments. For each style transfer task, we first generate 100 images in the target concept/style. We then extract feature representations at timestep $t = 10$ (out of 1000) from the conditional path at layers `up_blocks.0.resnets.0`, `up_blocks.0.resnets.1`, `up_blocks.0.resnets.2`, `up_blocks.1.resnets.0`, `up_blocks.1.resnets.1`, and `up_blocks.0.resnets.2`. The resulting tensor has size $100 \times C \times H \times W$. We compute the mean across the image, height, and width dimensions, yielding a steering vector of size $1 \times C \times 1 \times 1$. Representation steering is performed by adding this steering vector to the conditional path representation of a source image with varying editing

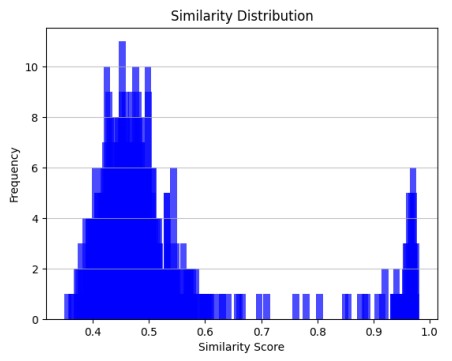

(a) Bimodal similarity of generated samples to the training set (CIFAR10) under duplication

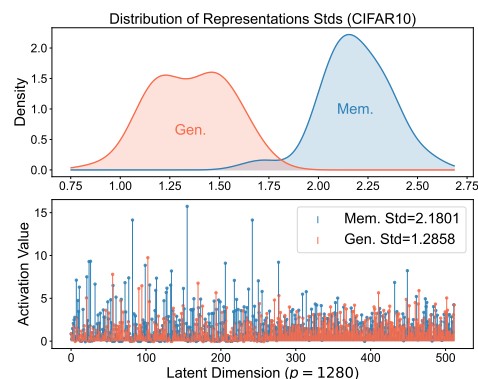

(b) Mem./Gen. representation statistics for an EDM pretrained on CIFAR10 with a duplicated subset.

Figure 16: Effect of training-set duplication. Duplicates induce a memorization mode while non-duplicated data continue to support generalization.

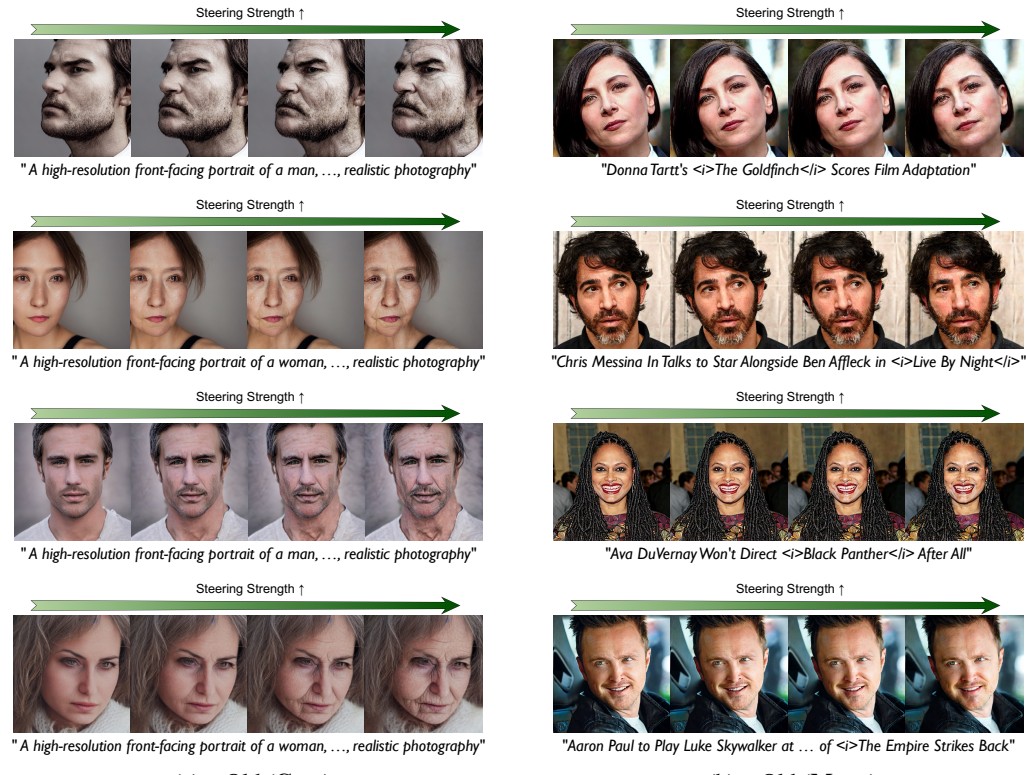

(a) +Old (**Gen.**)

(b) +Old (**Mem.**)

Figure 17: **Image editing via *single-layer* representation steering.** We follow the setup of Figure 7, but extract and apply the steering vector using only one layer.

strengths. Sampling is performed with 40 total generation steps, where representation steering is applied during the final 20 steps. All experiments use a classifier-free guidance (CFG) scale of 3.5.

## C.4 EXPLORATION ON STEERING-BASED IMAGE EDITING

In the main body of the paper, we show that a simple representation-based steering method enables effective image editing. More importantly, the editing outcomes differ systematically between generalized and memorized images. In this subsection, we evaluate the robustness of this method.

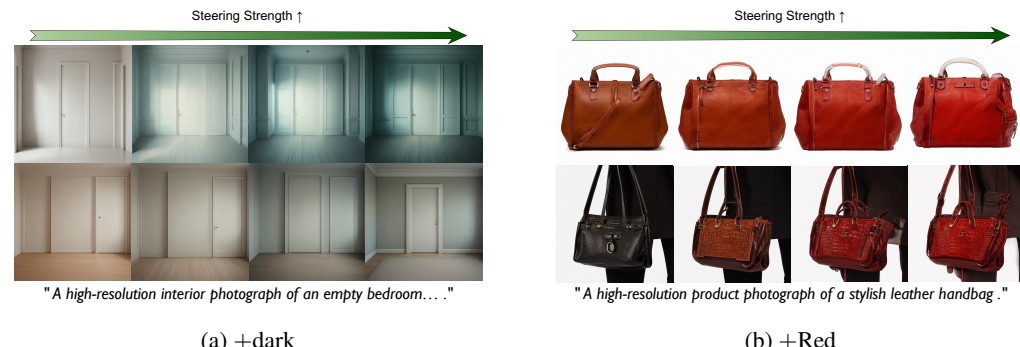

Figure 18: **Image editing on SD 3.5.** We follow the setup of Figure 7 using more recent DiT-based Stable Diffusion 3.5 model for image editing.

- **Using fewer layers.** As described in Appendix C.3, the steering results in Figure 7 are obtained by extracting and applying representation addition across 6 layers of the network. As an ablation, we find that the method does not require such depth: even a single layer is sufficient. To illustrate this, we use `up_blocks.1.resnets.1` for both extraction and application, and present the results in Figure 17. The outputs closely match those in the main paper, and the distinction between memorized and generalized examples remains evident.

- **Stable Diffusion 3.5.** The representation space in this architecture is more elusive, likely due to components such as Adaptive LayerNorm. To investigate this, we applied representation steering using layers `transformer_blocks.10.norm1`, `transformer_blocks.11.norm1`, `transformer_blocks.12.norm1`, `transformer_blocks.13.norm1`, and `transformer_blocks.14.norm1`. We generated 200 reference images to extract representations for each task. Sampling was performed with 40 total generation steps, with steering applied during steps 35–30. All experiments utilized a CFG scale of 4.5. These results are visualized in Figure 18.

We note that we do not intend to compete with existing outstanding steering methods, as several such approaches have already demonstrated impressive empirical success (Hertz et al., 2023; Zhang et al., 2023; Gandikota et al., 2024; Kadkhodaie et al., 2024b; Chen et al., 2024b). Rather, our focus is on showing how steering reveals the dual relationship between representation structure and generation behavior. In particular, when the model generalizes, its representations form compositional and interpretable spaces, enabling continuous and controllable edits.

## D DEFERRED PROOFS

### D.1 PROOF OF LEMMA D.1

**Lemma D.1** (Global minimizers of Regularized LAE). *Consider the regularized $p$-neuron LAE objective with $\boldsymbol{W}_1, \boldsymbol{W}_2 \in \mathbb{R}^{d \times p}$:*

$$\hat{\mathcal{L}}_{\boldsymbol{X}}(\boldsymbol{W}_2, \boldsymbol{W}_1) := \|\boldsymbol{W}_2 \boldsymbol{W}_1^\top \boldsymbol{X} - \boldsymbol{X}\|_F^2 + n\sigma^2 \|\boldsymbol{W}_2 \boldsymbol{W}_1^\top\|_F^2 + \lambda' \big(\|\boldsymbol{W}_1\|_F^2 + \|\boldsymbol{W}_2\|_F^2\big),$$

*where $\boldsymbol{X} = (\boldsymbol{x}_1, \ldots, \boldsymbol{x}_n)$ and $\boldsymbol{S} := \boldsymbol{X}\boldsymbol{X}^\top = \boldsymbol{U}\boldsymbol{\Lambda}\boldsymbol{U}^\top$. Assume $\lambda' < \lambda_p$, where $\lambda_1 \geq \cdots \geq \lambda_d$ are the eigenvalues of $\boldsymbol{S}$. Then every global minimizer has the form*

$$\boldsymbol{W}_2^\star = \boldsymbol{W}_1^\star = \boldsymbol{U}_{(p)} \big(\boldsymbol{I} + n\sigma^2 \boldsymbol{\Lambda}_{(p)}^{-1}\big)^{-\frac{1}{2}} \big(\boldsymbol{I} - \lambda' \boldsymbol{\Lambda}_{(p)}^{-1}\big)^{\frac{1}{2}} \boldsymbol{O}^\top := \boldsymbol{W}_{\boldsymbol{X}}, \tag{12}$$

*where $\boldsymbol{U}_{(p)}$ contains the top-$p$ eigenvectors, $\boldsymbol{\Lambda}_{(p)}$ the corresponding eigenvalues, and $\boldsymbol{O} \in \mathbb{R}^{p \times p}$ is any orthogonal matrix.*

*Proof.* **(0) Idea.** Set $\boldsymbol{A} = \boldsymbol{W}_2 \boldsymbol{W}_1^\top$ and replace the separate Frobenius penalties with a nuclear norm via $\min_{\boldsymbol{W}_1, \boldsymbol{W}_2: \boldsymbol{W}_2 \boldsymbol{W}_1^\top = \boldsymbol{A}}(\|\boldsymbol{W}_1\|_F^2 + \|\boldsymbol{W}_2\|_F^2) = 2\|\boldsymbol{A}\|_*$. Rotate to the $\boldsymbol{S}$-basis and pinch to

diagonalize, yielding $d$ decoupled 1D convex problems with solutions $\alpha_i^\star = \left(\frac{\lambda_i - \lambda'}{\lambda_i + n\sigma^2}\right)_+$. Keep the top $p$ directions (largest $\lambda_i$), then factor $A^\star$ optimally to obtain (12).

**(1) Reduction to a convex objective in $A = W_2 W_1^\top$.** For any $A$ with $\mathrm{rank}(A) \le p$,

$$\min_{W_1, W_2 : W_2 W_1^\top = A} \left( \|W_1\|_F^2 + \|W_2\|_F^2 \right) = 2\|A\|_*.$$

Hence

$$\min_{W_1, W_2} \hat{\mathcal{L}}_X(W_2, W_1) = \min_{\substack{A \in \mathbb{R}^{d \times d} \\ \mathrm{rank}(A) \le p}} \left( \|AX - X\|_F^2 + n\sigma^2 \|A\|_F^2 + 2\lambda' \|A\|_* \right) =: \min_A F(A),$$

where $F$ is convex in $A$ (the rank constraint is nonconvex).

**(2) Diagonalization in the $S$-basis.** Let $A = U\tilde{A}U^\top$ with $S = U\Lambda U^\top$. Using

$$\|AX - X\|_F^2 = \mathrm{Tr}(ASA^\top) - 2\,\mathrm{Tr}(AS) + \mathrm{Tr}(S),$$

we obtain

$$F(A) = \underbrace{\mathrm{Tr}(\tilde{A}\Lambda\tilde{A}^\top)}_{=\sum_j \lambda_j \sum_i \tilde{a}_{ij}^2} - 2\sum_i \lambda_i \tilde{a}_{ii} + n\sigma^2 \|\tilde{A}\|_F^2 + 2\lambda' \|\tilde{A}\|_* + \mathrm{Tr}(\Lambda).$$

Zeroing the off-diagonal entries of $\tilde{A}$ weakly decreases the quadratic terms and does not increase the nuclear norm (pinching (Bhatia, 2013)). Thus a minimizer can be chosen diagonal in the $U$-basis: $A = U \,\mathrm{diag}(\alpha_1, \ldots, \alpha_d)\, U^\top$.

**(3) Scalar decoupling and positivity.** With $A$ diagonal as above,

$$F(A) = \sum_{i=1}^d \left[ \lambda_i (1 - \alpha_i)^2 + n\sigma^2 \alpha_i^2 + 2\lambda' |\alpha_i| \right] + \text{const.}$$

For $\lambda_i \ge 0$, negatives are suboptimal (replacing $\alpha$ by $|\alpha|$ decreases the first term), so we minimize over $\alpha_i \ge 0$:

$$\alpha_i^\star = \left( \frac{\lambda_i - \lambda'}{\lambda_i + n\sigma^2} \right)_+.$$

**(4) Rank-$p$ constraint and form of the minimizer.** Enforcing $\mathrm{rank}(A) \le p$ keeps the $p$ indices with largest $\lambda_i$ (equivalently, largest unconstrained $\alpha_i^\star$) and sets the rest to 0. Writing $\alpha_i^\star = s_i^2$ on this set,

$$s_i = \left( 1 + n\sigma^2 \lambda_i^{-1} \right)^{-\frac{1}{2}} \left( 1 - \lambda' \lambda_i^{-1} \right)^{\frac{1}{2}},$$

and

$$W_2^\star = W_1^\star = U_{(p)} \,\mathrm{diag}(s_i)\, O^\top = U_{(p)} \left( I + n\sigma^2 \Lambda_{(p)}^{-1} \right)^{-\frac{1}{2}} \left( I - \lambda' \Lambda_{(p)}^{-1} \right)^{\frac{1}{2}} O^\top,$$

with any orthogonal $O \in \mathbb{R}^{p \times p}$. This matches (12). (All inverses/square-roots are taken entrywise on $\Lambda_{(p)}$.) $\qquad\square$

**Remark (large $\lambda'$ or degenerate $S$).** If some $\lambda_i \le \lambda'$ (including $\lambda_i = 0$), then the unconstrained coefficients $\alpha_i^\star = \left(\frac{\lambda_i - \lambda'}{\lambda_i + n\sigma^2}\right)_+$ vanish on those indices. In that case, keep the $p$ largest indices with $\lambda_i > \lambda'$ (the rank may drop below $p$ if fewer exist), and the same formulas apply entrywise on the retained eigenvalues; any remaining columns can be set to zero and $O$ is arbitrary.

## D.2   PROOF OF THEOREM 3.1

**Definition D.1** (($\alpha, \beta$)-Separability of Training Data)**.** *Suppose the training dataset $\mathcal{D}$ can be partitioned into $M$ clusters $X = [X_1, \ldots, X_M]$, where $X_k = [x_{k,1}, \ldots, x_{k,n_k}] \subseteq \mathbb{R}^d$ has mean $\bar{x}_k := \frac{1}{n_k} \sum_{j=1}^{n_k} x_{k,j}$. We say the dataset is ($\alpha, \beta$)-separable if, for some $\alpha \in (0, 1)$ and $\beta < 0$,*

$$\frac{\|x_{k,j} - \bar{x}_k\|_2}{\|\bar{x}_k\|_2} \le \alpha \quad \text{for all } k, j, \qquad \frac{\langle \bar{x}_k, \bar{x}_\ell \rangle}{\|\bar{x}_k\|_2 \, \|\bar{x}_\ell\|_2} \le \beta \quad \text{for all } k \ne \ell.$$

**Theorem D.2** (Restatement of Theorem 3.1). *Assume $(\alpha, \beta)$-separability with $\beta < 0$ and nonde-generate means $\min_k \|\bar{\boldsymbol{x}}_k\|_2 \geq b > 0$. Let $n = \sum_{k=1}^{M} n_k$ and define*

$$\boldsymbol{W}_2^\star = \boldsymbol{W}_1^\star = (\boldsymbol{W}_{\boldsymbol{X}_1} \quad \cdots \quad \boldsymbol{W}_{\boldsymbol{X}_M}).$$

*For each $k$, let $\boldsymbol{X}_k \boldsymbol{X}_k^\top = \boldsymbol{U}_k \boldsymbol{\Lambda}_k \boldsymbol{U}_k^\top$ be the eigen-decomposition and let $\boldsymbol{U}_k^{(p_k)}$ collect the top $p_k$ eigenvectors (with eigenvalues $\boldsymbol{\Lambda}_k^{(p_k)}$). Assume $n\lambda < \lambda_{\min}(\boldsymbol{\Lambda}_k^{(p_k)})$, so that the block solutions below are well-defined (real). Then there exist absolute constants $C, c > 0$ and a margin $\gamma > 0$ (defined explicitly below, depending only on $\alpha, \beta, b, \{p_k\}$, and the block scalings, and independent of the noise level) such that, for all*

$$(\boldsymbol{W}_2, \boldsymbol{W}_1) \in \mathcal{B}_\delta := \{ \|\boldsymbol{W}_2 - \boldsymbol{W}_2^\star\|_F + \|\boldsymbol{W}_1 - \boldsymbol{W}_1^\star\|_F \leq \delta \} \quad \text{and all } \sigma > 0,$$

*we can decompose the DAE loss into LAE losses introduced in Lemma D.1:*

$$\mathcal{L}_{\boldsymbol{X}}(\boldsymbol{W}_2, \boldsymbol{W}_1) = \frac{1}{n} \sum_{k=1}^{M} \hat{\mathcal{L}}_{\boldsymbol{X}_k}(\boldsymbol{W}_{2,(k)}, \boldsymbol{W}_{1,(k)}) + \varepsilon(\delta, \sigma, \gamma), \qquad \varepsilon(\delta, \sigma, \gamma) \leq C\left(\frac{\delta}{\gamma} + e^{-c\gamma^2/\sigma^2}\right),$$

$$(13)$$

*where $\hat{\mathcal{L}}_{\boldsymbol{X}_k}$ is the LAE objective in Lemma D.1 for cluster $k$ with noise weight $n_k\sigma^2$ and weight decay $\lambda' = n\lambda$. Moreover, each block is minimized by*

$$\boldsymbol{W}_{2,(k)}^\star = \boldsymbol{W}_{1,(k)}^\star = \boldsymbol{W}_{\boldsymbol{X}_k} := \boldsymbol{U}_k^{(p_k)}\big(\boldsymbol{I} + n_k\sigma^2 \boldsymbol{\Lambda}_k^{(p_k)\,-1}\big)^{-\frac{1}{2}}\big(\boldsymbol{I} - n\lambda \boldsymbol{\Lambda}_k^{(p_k)\,-1}\big)^{\frac{1}{2}} \boldsymbol{O}_k^\top,$$

*for some orthogonal $\boldsymbol{O}_k$. Consequently $(\boldsymbol{W}_2^\star, \boldsymbol{W}_1^\star)$ is a local minimizer.*

*Furthermore, the constructed minimizer is close to an actual minimizer: On $\mathcal{B}_\delta$, if we fix the ReLU masks to be those induced by $(\boldsymbol{W}_2^\star, \boldsymbol{W}_1^\star)$ at the center (see Steps (1)–(2) below), then the map*

$$(\boldsymbol{W}_2, \boldsymbol{W}_1) \mapsto \frac{1}{n} \sum_k \hat{\mathcal{L}}_{\boldsymbol{X}_k}$$

*is $m_0$-strongly convex around $(\boldsymbol{W}_2^\star, \boldsymbol{W}_1^\star)$ with*

$$m_0 \geq c_0(\sigma^2 + n\lambda),$$

*for a numerical constant $c_0 > 0$ independent of $(\delta, \sigma, \gamma)$. Therefore, any local minimizer $(\widehat{\boldsymbol{W}}_2, \widehat{\boldsymbol{W}}_1)$ of the full DAE loss inside $\mathcal{B}_\delta$ obeys*

$$\big\|(\widehat{\boldsymbol{W}}_2, \widehat{\boldsymbol{W}}_1) - (\boldsymbol{W}_2^\star, \boldsymbol{W}_1^\star)\big\|_F \leq \sqrt{\frac{2\,\varepsilon(\delta, \sigma, \gamma)}{m_0}} \leq \sqrt{\frac{2C}{c_0}}\,\big(\sigma^2 + n\lambda\big)^{-1/2}\left(\frac{\delta}{\gamma} + e^{-c\gamma^2/\sigma^2}\right)^{1/2}.$$

$$(14)$$

*In particular, if $\delta/\gamma \to 0$ and $\sigma/\gamma \to 0$ (with $n\lambda > 0$ fixed), the right-hand side is $o(1)$, giving an explicit $o(1)$ control on the distance between the constructed solution and the actual local minimizer.*

*Proof.* Let $f_{\boldsymbol{W}_2, \boldsymbol{W}_1}(\boldsymbol{z}) = \boldsymbol{W}_2[\boldsymbol{W}_1^\top \boldsymbol{z}]_+$ and write $\boldsymbol{W}_1^\star = [\boldsymbol{W}_{\boldsymbol{X}_1}, \ldots, \boldsymbol{W}_{\boldsymbol{X}_M}]$, so the columns are partitioned into $M$ blocks. We also use $[\boldsymbol{v}]_- := [-\boldsymbol{v}]_+$ entrywise.

**(0) Idea.** We compare the nonlinear DAE $f_{\boldsymbol{W}_2, \boldsymbol{W}_1}$ with its mask-fixed linearized counterpart $f_{\boldsymbol{W}_2, \boldsymbol{W}_1}^{\mathrm{LAE}}$. The intended behavior is: block $k$ is active on $\boldsymbol{X}_k$ with a positive margin, while all other blocks are inactive with a negative margin. For small $(\sigma, \delta)$, the ReLU masks are preserved with high probability; concretely,

$$\boldsymbol{f}_{\boldsymbol{W}_2, \boldsymbol{W}_1}(\boldsymbol{X} + \sigma\boldsymbol{\varepsilon}) = \boldsymbol{W}_2\big[\boldsymbol{W}_1^\top(\boldsymbol{X}_1 + \sigma\boldsymbol{\varepsilon}_1, \ldots, \boldsymbol{X}_M + \sigma\boldsymbol{\varepsilon}_M)\big]_+$$

$$\approx (\boldsymbol{W}_{2,(1)} \quad \cdots \quad \boldsymbol{W}_{2,(M)}) \begin{pmatrix} [\boldsymbol{W}_{1,(1)}^\top(\boldsymbol{X}_1 + \sigma\boldsymbol{\varepsilon}_1)]_+ & & \\ & \ddots & \\ & & [\boldsymbol{W}_{1,(M)}^\top(\boldsymbol{X}_M + \sigma\boldsymbol{\varepsilon}_M)]_+ \end{pmatrix}$$

$$(15)$$

$$:= \boldsymbol{f}_{\boldsymbol{W}_2, \boldsymbol{W}_1}^{\mathrm{LAE}}(\boldsymbol{X} + \sigma\boldsymbol{\varepsilon}).$$

With masks fixed to the "correct" ones (block $k$ on $\boldsymbol{X}_k$, others off), the network reduces to a linear map

$$f^{\mathrm{LAE}}_{\boldsymbol{W}_2, \boldsymbol{W}_1}(\boldsymbol{z}) = \boldsymbol{W}_{2,(k)} \boldsymbol{W}^{\top}_{1,(k)} \boldsymbol{z} \qquad (\boldsymbol{z} \in \boldsymbol{X}_k),$$

i.e., each cluster is reconstructed by *a small number of neurons in its corresponding block*. Equivalently, writing $\boldsymbol{A}_k := \boldsymbol{W}_{2,(k)} \boldsymbol{W}^{\top}_{1,(k)}$ and $\boldsymbol{A} := \mathrm{blkdiag}(\boldsymbol{A}_1, \ldots, \boldsymbol{A}_M)$, we have $f^{\mathrm{LAE}}_{\boldsymbol{W}_2, \boldsymbol{W}_1}(\boldsymbol{X} + \sigma \boldsymbol{\varepsilon}) = \boldsymbol{A}(\boldsymbol{X} + \sigma \boldsymbol{\varepsilon})$. With fixed masks, the loss decouples into $M$ LAE problems and becomes solvable.

**(1) Masks and margins at the block center (no noise).** Write $\boldsymbol{W}_{\boldsymbol{X}_k} = \boldsymbol{U}^{(p_k)}_k \boldsymbol{S}_k \boldsymbol{O}^{\top}_k$[2] with $\boldsymbol{S}_k = \mathrm{diag}(s_{k,1}, \ldots, s_{k,p_k}) \succ 0$. Let $s_{\min} := \min_{k,r} s_{k,r}$ and $s_{\max} := \max_{k,r} s_{k,r}$, and choose $\boldsymbol{O}_k$ so that

$$\boldsymbol{O}^{\top}_k \boldsymbol{U}^{(p_k)\top}_k \bar{\boldsymbol{x}}_k = \frac{\|\boldsymbol{U}^{(p_k)\top}_k \bar{\boldsymbol{x}}_k\|_2}{\sqrt{p_k}} \mathbf{1}_{p_k}.$$

Since

$$\boldsymbol{X}_k \boldsymbol{X}^{\top}_k = n_k \, \bar{\boldsymbol{x}}_k \bar{\boldsymbol{x}}^{\top}_k + \sum_t (\boldsymbol{x}_{k,t} - \bar{\boldsymbol{x}}_k)(\boldsymbol{x}_{k,t} - \bar{\boldsymbol{x}}_k)^{\top},$$

the within-cluster tightness ($\alpha$) implies the "residual" term has spectral norm

$$\left\| \sum_t (\boldsymbol{x}_{k,t} - \bar{\boldsymbol{x}}_k)(\boldsymbol{x}_{k,t} - \bar{\boldsymbol{x}}_k)^{\top} \right\|_{op} \leq \sum_t \|\boldsymbol{x}_{k,t} - \bar{\boldsymbol{x}}_k\|^2_2 \leq n_k \alpha^2 \|\bar{\boldsymbol{x}}_k\|^2_2.$$

Therefore $\bar{\boldsymbol{x}}_k$ is well aligned with the top eigenspace of $\boldsymbol{X}_k \boldsymbol{X}^{\top}_k$. In particular, there exists $c_{\mathrm{proj}}(\alpha) \in (0, 1]$ such that

$$\frac{\|\boldsymbol{U}^{(p_k)\top}_k \bar{\boldsymbol{x}}_k\|_2}{\|\bar{\boldsymbol{x}}_k\|_2} \geq c_{\mathrm{proj}}(\alpha).$$

*(One explicit choice.)* Let $\boldsymbol{u}_{k,1}$ be the top eigenvector of $\boldsymbol{X}_k \boldsymbol{X}^{\top}_k$ (so $\boldsymbol{u}_{k,1} \in \mathrm{span}(\boldsymbol{U}^{(p_k)}_k)$ for any $p_k \geq 1$). A Davis–Kahan/Wedin-type bound gives

$$\sin \angle(\boldsymbol{u}_{k,1}, \bar{\boldsymbol{x}}_k) \leq \frac{\alpha^2}{1 - \alpha^2} \qquad \Longrightarrow \qquad \frac{|\langle \boldsymbol{u}_{k,1}, \bar{\boldsymbol{x}}_k \rangle|}{\|\bar{\boldsymbol{x}}_k\|_2} \geq \sqrt{1 - \left( \frac{\alpha^2}{1 - \alpha^2} \right)^2},$$

hence one may take $c_{\mathrm{proj}}(\alpha) := \sqrt{1 - (\alpha^2/(1 - \alpha^2))^2}$ (for $\alpha < 1$).

Hence, for any $\boldsymbol{x} \in \boldsymbol{X}_k$ and any column $\boldsymbol{w}^{\star}_{k,r}$ of $\boldsymbol{W}^{\star}_{1,(k)}$,

$$\langle \boldsymbol{w}^{\star}_{k,r}, \boldsymbol{x} \rangle \geq \|\bar{\boldsymbol{x}}_k\|_2 \left( s_{\min} \frac{c_{\mathrm{proj}}(\alpha)}{\sqrt{p_k}} - s_{\max} \alpha \right).$$

For any $\ell \neq k$ and any unit vector $\boldsymbol{u} \in \mathrm{span}(\boldsymbol{U}^{(p_\ell)}_\ell)$, $(\alpha, \beta)$-separability yields $\langle \boldsymbol{u}, \bar{\boldsymbol{x}}_k \rangle \leq \beta + \alpha$. Therefore, for any column $\boldsymbol{w}^{\star}_{\ell,r}$,

$$\langle \boldsymbol{w}^{\star}_{\ell,r}, \boldsymbol{x} \rangle \leq \|\bar{\boldsymbol{x}}_k\|_2 s_{\max}(\beta + 2\alpha) \leq -\|\bar{\boldsymbol{x}}_k\|_2 s_{\max} \frac{|\beta|}{2},$$

provided $\alpha$ is sufficiently small compared to $|\beta|$ (absorbed into constants below). Define the *margin*

$$\gamma := \min_k \|\bar{\boldsymbol{x}}_k\|_2 \cdot \min \left\{ s_{\min} \frac{c_{\mathrm{proj}}(\alpha)}{\sqrt{p_k}} - s_{\max} \alpha , \ \frac{s_{\max}|\beta|}{2} \right\} > 0. \qquad (16)$$

Then, on $\boldsymbol{X}_k$, every unit in block $k$ has pre-activation $\geq \gamma$ and every unit in $\ell \neq k$ has pre-activation $\leq -\gamma$.

**(2) Mask stability with noise $\boldsymbol{\varepsilon}$, and loss error.** Fix a noise draw $\boldsymbol{\varepsilon} = (\boldsymbol{\varepsilon}_1, \ldots, \boldsymbol{\varepsilon}_n)$ and set

$$\boldsymbol{e}(\boldsymbol{\varepsilon}) := f_{\boldsymbol{W}_2, \boldsymbol{W}_1}(\boldsymbol{X} + \sigma \boldsymbol{\varepsilon}) - f^{\mathrm{LAE}}_{\boldsymbol{W}_2, \boldsymbol{W}_1}(\boldsymbol{X} + \sigma \boldsymbol{\varepsilon}) = [\boldsymbol{e}_1 \ \cdots \ \boldsymbol{e}_n] \in \mathbb{R}^{d \times n},$$

---

[2] Any right-orthogonal choice of $\boldsymbol{O}_k$ yields the same objective value; the choice above maximizes the first-order margin and is convenient for the mask analysis.

where, for a sample $\boldsymbol{x}_{i_k}$ from the $i$-th cluster, the deviation is

$$\boldsymbol{e}_{i_k} = \sum_{\ell \neq i} \boldsymbol{W}_{2,(\ell)} \underbrace{\left[\boldsymbol{W}_{1,(\ell)}^{\top}(\boldsymbol{x}_{i_k} + \sigma\boldsymbol{\varepsilon}_{i_k})\right]_+}_{\text{off-block, should be 0}} - \boldsymbol{W}_{2,(i)} \underbrace{\left[\boldsymbol{W}_{1,(i)}^{\top}(\boldsymbol{x}_{i_k} + \sigma\boldsymbol{\varepsilon}_{i_k})\right]_-}_{\text{on-block, should be 0}}. \qquad (17)$$

Intuitively, $\boldsymbol{e}_{i_k} = \boldsymbol{0}$ unless some pre-activation crosses the margin. Moreover,

$$\|\boldsymbol{e}(\boldsymbol{\varepsilon})\|_F^2 = \sum_{j=1}^n \|\boldsymbol{e}_j\|_2^2.$$

**Loss difference.** For a fixed noise realization $\boldsymbol{\varepsilon}$, define

$$\mathcal{L}_{\boldsymbol{\varepsilon}}(\boldsymbol{W}_2, \boldsymbol{W}_1) := \frac{1}{n}\big\|f_{\boldsymbol{W}_2,\boldsymbol{W}_1}(\boldsymbol{X} + \sigma\boldsymbol{\varepsilon}) - \boldsymbol{X}\big\|_F^2 \ + \ \lambda\big(\|\boldsymbol{W}_1\|_F^2 + \|\boldsymbol{W}_2\|_F^2\big),$$

and analogously $\mathcal{L}_{\boldsymbol{\varepsilon}}^{\mathrm{LAE}}$ with $f^{\mathrm{LAE}}$. Writing $\boldsymbol{a} := f_{\boldsymbol{W}_2,\boldsymbol{W}_1}(\boldsymbol{X} + \sigma\boldsymbol{\varepsilon}) - \boldsymbol{X}$ and $\boldsymbol{b} := f_{\boldsymbol{W}_2,\boldsymbol{W}_1}^{\mathrm{LAE}}(\boldsymbol{X} + \sigma\boldsymbol{\varepsilon}) - \boldsymbol{X}$ (so $\boldsymbol{a} - \boldsymbol{b} = \boldsymbol{e}(\boldsymbol{\varepsilon})$),

$$\begin{aligned}
\big|\mathcal{L}_{\boldsymbol{\varepsilon}} - \mathcal{L}_{\boldsymbol{\varepsilon}}^{\mathrm{LAE}}\big| &= \frac{1}{n}\left|\|\boldsymbol{a}\|_F^2 - \|\boldsymbol{b}\|_F^2\right| = \frac{1}{n}\left|\langle \boldsymbol{a} + \boldsymbol{b}, \, \boldsymbol{a} - \boldsymbol{b}\rangle\right| \\
&\leq \frac{1}{n}\big(\|\boldsymbol{a}\|_F + \|\boldsymbol{b}\|_F\big)\,\|\boldsymbol{e}(\boldsymbol{\varepsilon})\|_F \\
&\leq \frac{1}{n}\Big(2\|\boldsymbol{b}\|_F + \|\boldsymbol{e}(\boldsymbol{\varepsilon})\|_F\Big)\,\|\boldsymbol{e}(\boldsymbol{\varepsilon})\|_F,
\end{aligned}$$

where the last line uses $\|\boldsymbol{a}\|_F \leq \|\boldsymbol{b}\|_F + \|\boldsymbol{e}(\boldsymbol{\varepsilon})\|_F$. Thus it remains to bound $\mathbb{E}\|\boldsymbol{e}(\boldsymbol{\varepsilon})\|_F^2$ and $\mathbb{E}\|\boldsymbol{b}\|_F$.

We further simplify $\|\boldsymbol{b}\|_F$ by splitting out the noise. Denote $\boldsymbol{A}_k := \boldsymbol{W}_{2,(k)}\boldsymbol{W}_{1,(k)}^{\top}$ and $\boldsymbol{A} :=$ blkdiag$(\boldsymbol{A}_1, \ldots, \boldsymbol{A}_M)$, so

$$\|f_{\boldsymbol{W}_2,\boldsymbol{W}_1}^{\mathrm{LAE}}(\boldsymbol{X} + \sigma\boldsymbol{\varepsilon}) - \boldsymbol{X}\|_F = \|\boldsymbol{A}(\boldsymbol{X} + \sigma\boldsymbol{\varepsilon}) - \boldsymbol{X}\|_F \leq \|(\boldsymbol{A} - \boldsymbol{I})\boldsymbol{X}\|_F + \sigma\,\|\boldsymbol{A}\|_{op}\,\|\boldsymbol{\varepsilon}\|_F.$$

Taking expectations and using Cauchy–Schwarz with $\mathbb{E}\|\boldsymbol{\varepsilon}\|_F^2 = dn$ reduces the problem to bounding $\mathbb{E}\|\boldsymbol{e}(\boldsymbol{\varepsilon})\|_F^2$.

**Bounding $\mathbb{E}\|\boldsymbol{e}(\boldsymbol{\varepsilon})\|_F^2$.** Fix a column $\boldsymbol{e}_{i_k}$. The entries in $\left[\boldsymbol{W}_{1,(\ell)}^{\top}(\boldsymbol{x}_{i_k} + \sigma\boldsymbol{\varepsilon}_{i_k})\right]_+$ are rectified Gaussians. By the margin argument in Step (1), at the *center* $(\boldsymbol{W}_2^{\star}, \boldsymbol{W}_1^{\star})$ we have $\boldsymbol{W}_{1,(\ell)}^{\star\top}\boldsymbol{x}_{i_k} \preceq -\gamma\boldsymbol{1}$ for $\ell \neq i$ and $\boldsymbol{W}_{1,(i)}^{\star\top}\boldsymbol{x}_{i_k} \succeq +\gamma\boldsymbol{1}$.[3]

Thus it suffices to control the first and second moments of a rectified Gaussian whose mean is separated from 0 by $\gamma$. Let $Z \sim \mathcal{N}(\mu, s^2)$ with $\mu \leq -\gamma$. A standard Gaussian tail bound (Mills ratio / Chernoff) implies

$$\mathbb{E}[Z]_+ \leq \frac{s^2}{-\mu}\,e^{-\mu^2/(2s^2)} \leq \frac{s^2}{\gamma}\,e^{-\gamma^2/(2s^2)}, \qquad \mathbb{E}[Z]_+^2 \leq \frac{s^4}{\mu^2}\,e^{-\mu^2/s^2} \leq \frac{s^4}{\gamma^2}\,e^{-\gamma^2/s^2},$$

and the same bounds hold for within-block terms with $[Z]_- = [-Z]_+$.

Now, a generic off-block pre-activation (an entry in (17)) has the form

$$Z = \boldsymbol{w}^{\top}\boldsymbol{x} + \sigma\boldsymbol{w}^{\top}\boldsymbol{\varepsilon}, \qquad \mathbb{E}Z = \mu \leq -\gamma, \quad \mathrm{Var}(Z) = s^2 = \sigma^2\|\boldsymbol{w}\|_2^2.$$

Let

$$\kappa := \max_r \|\boldsymbol{w}_{1,r}\|_2, \qquad L_2^2 := \sum_{j=1}^M \|\boldsymbol{W}_{2,(j)}\|_{op}^2, \qquad p := \sum_{j=1}^M p_j.$$

Using $\|\boldsymbol{W}_{2,(\ell)}\boldsymbol{v}\|_2 \leq \|\boldsymbol{W}_{2,(\ell)}\|_{op}\|\boldsymbol{v}\|_2$ and the second-moment bound above,

$$\mathbb{E}\|\boldsymbol{e}_{i_k}\|_2^2 \ \leq \ \frac{\sigma^4\kappa^4}{\gamma^2}\,e^{-\gamma^2/(\sigma^2\kappa^2)}\,L_2^2\,p.$$

---

[3]On $\mathcal{B}_\delta$, the pre-activations shift by at most $O(\delta)$ (absorbed into the $\delta/\gamma$ term in the final bound), so we may treat the mean as $\leq -\gamma$ (off-block) or $\geq +\gamma$ (on-block) up to constants.

Since $\|\boldsymbol{e}(\boldsymbol{\varepsilon})\|_F^2 = \sum_{j=1}^n \|\boldsymbol{e}_j\|_2^2$, we obtain

$$\mathbb{E}\,\|\boldsymbol{e}(\boldsymbol{\varepsilon})\|_F^2 \;=\; \sum_{j=1}^n \mathbb{E}\,\|\boldsymbol{e}_j\|_2^2 \;\leq\; \frac{\sigma^4 \kappa^4}{\gamma^2}\, e^{-\gamma^2/(\sigma^2 \kappa^2)}\, n\, L_2^2\, p.$$

**Final plug-in.** Let $\boldsymbol{A} := \mathrm{blkdiag}(\boldsymbol{A}_1, \ldots, \boldsymbol{A}_M)$, $L_A := \|\boldsymbol{A}\|_{op}$, and $B_{\mathrm{LAE}} := \|(\boldsymbol{A} - \boldsymbol{I})\boldsymbol{X}\|_F$. The preceding bounds imply

$$\left|\mathcal{L}_{\boldsymbol{X}}(\boldsymbol{W}_2, \boldsymbol{W}_1) - \mathcal{L}_{\boldsymbol{X}}^{\mathrm{LAE}}(\boldsymbol{W}_2, \boldsymbol{W}_1)\right| \;\leq\; C\!\left(\frac{\delta}{\gamma} \;+\; e^{-c\,\gamma^2/\sigma^2}\right)$$

uniformly on $\mathcal{B}_\delta$, for some absolute constants $C, c > 0$. (Here $\delta/\gamma$ accounts for deterministic mask changes across $\mathcal{B}_\delta$, while the exponential term accounts for noise-induced sign flips.)

**(3) Expectation yields the** LAE **loss.** For $\boldsymbol{\varepsilon} \sim \mathcal{N}(0, \boldsymbol{I})$ and $\boldsymbol{A}_k := \boldsymbol{W}_{2,(k)}\boldsymbol{W}_{1,(k)}^\top$,

$$\mathbb{E}\big\|\boldsymbol{A}_k(\boldsymbol{x} + \sigma\boldsymbol{\varepsilon}) - \boldsymbol{x}\big\|_2^2 = \|\boldsymbol{A}_k \boldsymbol{x} - \boldsymbol{x}\|_2^2 + \sigma^2 \|\boldsymbol{A}_k\|_F^2.$$

Summing over $\boldsymbol{x} \in \boldsymbol{X}_k$, averaging by $n$, and adding weight decay gives

$$\mathcal{L}_{\boldsymbol{X}}^{\mathrm{LAE}}(\boldsymbol{W}_2, \boldsymbol{W}_1) = \mathbb{E}_{\boldsymbol{\varepsilon}}\big[\mathcal{L}_{\boldsymbol{\varepsilon}}^{\mathrm{LAE}}\big] = \frac{1}{n}\sum_{k=1}^M \hat{\mathcal{L}}_{\boldsymbol{X}_k}(\boldsymbol{W}_{2,(k)}, \boldsymbol{W}_{1,(k)}),$$

using $\lambda' = n\lambda$ and $\sum_k \|\boldsymbol{W}_{i,(k)}\|_F^2 = \|\boldsymbol{W}_i\|_F^2$ for $i = 1, 2$.

**(4) Block solutions and distance to a strict minimizer.** By Lemma D.1, each block is minimized by $\boldsymbol{W}_{2,(k)}^\star = \boldsymbol{W}_{1,(k)}^\star = \boldsymbol{W}_{\boldsymbol{X}_k}$; concatenating blocks yields $(\boldsymbol{W}_2^\star, \boldsymbol{W}_1^\star)$, which minimize the leading LAE term in (13). The leading term $\frac{1}{n}\sum_k \hat{\mathcal{L}}_{\boldsymbol{X}_k}$ is quadratic in $(\boldsymbol{W}_2, \boldsymbol{W}_1)$ on $\mathcal{B}_\delta$ (with masks fixed). Its Hessian at $(\boldsymbol{W}_2^\star, \boldsymbol{W}_1^\star)$ equals the Hessian of the quadratic reconstruction term plus $2n\lambda\,\boldsymbol{I}$ from weight decay, together with the positive-semidefinite curvature from $\sigma^2 \|\boldsymbol{A}_k\|_F^2$. By continuity of the Hessian, this yields a uniform lower bound

$$\nabla^2\!\Big(\tfrac{1}{n}\sum_k \hat{\mathcal{L}}_{\boldsymbol{X}_k}\Big) \;\succeq\; m_0\,\boldsymbol{I} \quad \text{on a neighborhood of } (\boldsymbol{W}_2^\star, \boldsymbol{W}_1^\star), \qquad m_0 \;\geq\; c_0(\sigma^2 + n\lambda),$$

for a numerical $c_0 > 0$ independent of $(\delta, \sigma, \gamma)$. Hence, for any local minimizer $(\widehat{\boldsymbol{W}}_2, \widehat{\boldsymbol{W}}_1)$ of the full DAE loss in $\mathcal{B}_\delta$,

$$\frac{m_0}{2}\big\|(\widehat{\boldsymbol{W}}_2, \widehat{\boldsymbol{W}}_1) - (\boldsymbol{W}_2^\star, \boldsymbol{W}_1^\star)\big\|_F^2 \;\leq\; \mathcal{L}_{\boldsymbol{X}}(\widehat{\boldsymbol{W}}_2, \widehat{\boldsymbol{W}}_1) - \mathcal{L}_{\boldsymbol{X}}(\boldsymbol{W}_2^\star, \boldsymbol{W}_1^\star) \;\leq\; \varepsilon(\delta, \sigma, \gamma),$$

which yields (14). The right-hand side is $o(1)$ whenever $\delta/\gamma \to 0$ and $\sigma/\gamma \to 0$, completing the proof. $\qquad\square$

### D.3 Proof of Corollary 3.2

**Corollary D.3** (Restatement of Cor. 3.2)**.** *Assume the dataset* $X = [\boldsymbol{x}_i \ldots, \boldsymbol{x}_n] \subset \mathbb{R}^d$ *is satisfy the separability condition in Definition 3.1. Consider an overparameterized ReLU DAE with* $p \geq n$ *hidden units, weight decay* $\lambda \geq 0$*, and input noise level* $\sigma > 0$*. Then, by Theorem 3.1 (applied to singleton clusters), there exists a local minimizer of the form*

$$\boldsymbol{W}_2^\star = \boldsymbol{W}_1^\star = (r_1 \boldsymbol{x}_1 \quad \cdots \quad r_n \boldsymbol{x}_n \quad \boldsymbol{0} \quad \cdots \quad \boldsymbol{0}) =: \boldsymbol{W}_{\mathrm{mem}}, \qquad r_i = \sqrt{\frac{\|\boldsymbol{x}_i\|_2^2 - n\lambda}{\|\boldsymbol{x}_i\|_2^4 + \sigma^2 \|\boldsymbol{x}_i\|_2^2}}.$$
$$(18)$$

*(The trailing* $(p - n)$ *columns are zero; see also Corollary D.4 on* $\ell_\infty$*-smoothness.) Moreover, for* $\lambda \to 0$ *this solution attains a small empirical loss independent of* $d$:

$$\mathcal{L}_{\boldsymbol{x}_i}(\boldsymbol{W}_{\mathrm{mem}}, \boldsymbol{W}_{\mathrm{mem}}) \;\lesssim\; \frac{\sigma^2 \|\boldsymbol{x}_i\|_2^2}{\sigma^2 + \|\boldsymbol{x}_i\|_2^2} \;<\; \sigma^2, \forall 1 \leq i \leq n$$

*Proof.* **(0) Optimal weights align with data.** Apply Theorem 3.1 with clusters $\boldsymbol{X}_k = \{\boldsymbol{x}_k\}$ of size $n_k = 1$. The block solution from Thm. 3.1 now yields $\boldsymbol{W}_{\boldsymbol{X}_k}^\star = r_k \frac{\boldsymbol{x}_k}{\|\boldsymbol{x}_k\|_2}$ with $r_k = \sqrt{\frac{\|\boldsymbol{x}_k\|_2^2 - n\lambda}{\|\boldsymbol{x}_k\|_2^2 + \sigma^2}}$, so the corresponding column of $\boldsymbol{W}_1^\star$ (and $\boldsymbol{W}_2^\star$) equals $r_k \boldsymbol{x}_k$ with $r_k = s_k / \|\boldsymbol{x}_k\|_2$, giving (18). Since $p \geq n$, we may set the remaining $(p - n)$ columns to zero without affecting the network output. Other equivalent parametrizations (e.g., duplicating columns and rescaling) have larger $\ell_\infty$-smoothness; our choice is the sparsest among these.

**(1) Empirical loss bound (case $\lambda \to 0$).** By Theorem 3.1 and Lemma D.1, with singleton clusters and $\lambda = 0$, the expected denoising loss decouples over samples and, for each $i$,

$$\min_{\alpha \in [0,1]} \left[ (1 - \alpha)^2 \|\boldsymbol{x}_i\|_2^2 + \sigma^2 \alpha^2 \right] = \frac{\sigma^2 \|\boldsymbol{x}_i\|_2^2}{\sigma^2 + \|\boldsymbol{x}_i\|_2^2}, \quad \text{attained at } \alpha_i^\star = \frac{\|\boldsymbol{x}_i\|_2^2}{\|\boldsymbol{x}_i\|_2^2 + \sigma^2}.$$

Averaging over $i$ gives the stated bound, which is strictly less than $\sigma^2$ and independent of the ambient dimension $d$. $\square$

## D.4 PROOF OF SMOOTHNESS WITH RESPECT TO THE INFINITY NORM

**Corollary D.4** (Sparse solution has the smoothest $\ell_\infty$ local landscape). *At the memorized, sparse solution $\boldsymbol{W}_2 = \boldsymbol{W}_1 = \boldsymbol{W}_{\text{mem}}$ from Corollary 3.2, the loss decomposes over singleton clusters as*

$$\sum_{i=1}^n \hat{\mathcal{L}}_{\boldsymbol{x}_i}(\boldsymbol{W}_2, \boldsymbol{W}_1) = \|A_i \boldsymbol{x}_i - \boldsymbol{x}_i\|_2^2 + \sigma^2 \|A_i\|_F^2 + 2n\lambda r_i^2 \|\boldsymbol{x}_i\|_2^2, \qquad A_i := \boldsymbol{W}_2 \boldsymbol{W}_1^\top = r_i^2 \boldsymbol{x}_i \boldsymbol{x}_i^\top.$$

*With masks frozen (singleton case), the Hessian w.r.t. $\boldsymbol{W}_1$ is block diagonal:*

$$\nabla_{\boldsymbol{W}_1}^2 \mathcal{L}_{\boldsymbol{X}}(\boldsymbol{W}_2, \boldsymbol{W}_1) = \text{blkdiag}\Big( \boldsymbol{H}(\boldsymbol{x}_1) + \lambda \boldsymbol{I}, \ldots, \boldsymbol{H}(\boldsymbol{x}_n) + \lambda \boldsymbol{I}, \underbrace{\lambda \boldsymbol{I}, \ldots, \lambda \boldsymbol{I}}_{p-n \text{ blocks}} \Big),$$

*where each active block has rank-1 plus diagonal form*

$$\boldsymbol{H}(\boldsymbol{x}_i) = a_i \boldsymbol{x}_i \boldsymbol{x}_i^\top, \qquad a_i > 0 \text{ (smooth in } \sigma, \lambda, r_i, \|\boldsymbol{x}_i\|_2).$$

*Consequently, the $\ell_\infty$ Lipschitz constant of the gradient at $\boldsymbol{W}_{\text{mem}}$ is*

$$L_\infty = \big\| \nabla_{\boldsymbol{W}_1}^2 \mathcal{L}_{\boldsymbol{X}}(\boldsymbol{W}_{\text{mem}}, \boldsymbol{W}_{\text{mem}}) \big\|_{\infty \to \infty} = \max\Big\{ \max_{1 \leq i \leq n} \|\boldsymbol{H}(\boldsymbol{x}_i) + \lambda \boldsymbol{I}\|_{\infty \to \infty}, \lambda \Big\}.$$

*Among all equivalent local minima obtained by orthogonal re-mixing within the active span (i.e., $\boldsymbol{W}_1 \mapsto \boldsymbol{W}_1 \boldsymbol{Q}$ and $\boldsymbol{W}_2 \mapsto \boldsymbol{W}_2 \boldsymbol{Q}$ with block-orthogonal $\boldsymbol{Q}$ that preserves masks and the LAE optimum), the choice $\boldsymbol{W}_{\text{mem}}$ minimizes $L_\infty$ and hence yields the smoothest local landscape in $\ell_\infty$.*

*Proof.* **(1) Block structure.** Freezing masks at singleton clusters forces second-order decoupling across columns, giving the block-diagonal Hessian displayed above. A direct differentiation of the decomposed objective shows each active block equals $a_i \boldsymbol{x}_i \boldsymbol{x}_i^\top + \lambda \boldsymbol{I}$ for some $a_i > 0$.

**(2) Bound $\ell_\infty$ via a $(1,1)$-norm of Hessian.** Recall $\|\boldsymbol{M}\|_{\infty \to \infty} = \max_{\|\boldsymbol{u}\|_\infty = 1} \|\boldsymbol{M}\boldsymbol{u}\|_\infty = \|\boldsymbol{M}^\top\|_{1 \to 1}$; for symmetric blocks this equals the maximum absolute column sum. Since the Hessian is block diagonal,

$$\big\| \nabla_{\boldsymbol{W}_1}^2 \mathcal{L}_{\boldsymbol{X}} \big\|_{\infty \to \infty} = \max\Big\{ \max_i \|a_i \boldsymbol{x}_i \boldsymbol{x}_i^\top + \lambda \boldsymbol{I}\|_{1 \to 1}, \lambda \Big\}.$$

For a rank-1 matrix, the $(1,1)$ operator norm is the max column sum:

$$\big\| a_i \boldsymbol{x}_i \boldsymbol{x}_i^\top \big\|_{1 \to 1} = a_i \|\boldsymbol{x}_i\|_1 \|\boldsymbol{x}_i\|_\infty.$$

Adding $\lambda \boldsymbol{I}$ increases each (absolute) column sum by at most $\lambda$, hence

$$\big\| a_i \boldsymbol{x}_i \boldsymbol{x}_i^\top + \lambda \boldsymbol{I} \big\|_{\infty \to \infty} = \big\| a_i \boldsymbol{x}_i \boldsymbol{x}_i^\top + \lambda \boldsymbol{I} \big\|_{1 \to 1} \leq a_i \|\boldsymbol{x}_i\|_1 \|\boldsymbol{x}_i\|_\infty + \lambda,$$

which yields the claimed expression for $L_\infty$.

**(3) Minimality under orthogonal re-mixing.** Let an equivalent optimum be obtained by block-orthogonal mixing that preserves masks. Each active block is conjugated to $\boldsymbol{Q}_i^\top (a_i \boldsymbol{x}_i \boldsymbol{x}_i^\top + \lambda \boldsymbol{I}) \boldsymbol{Q}_i$. While eigenvalues are invariant, $(1,1)$ (hence $\infty \to \infty$) norms are sensitive to *densification*. The memorized alignment keeps the block *rank-1 plus diagonal along $\boldsymbol{x}_i$*, which minimizes absolute column/row sums; mixing spreads mass across coordinates and (weakly) increases the max column/row sum. Therefore the memorized choice minimizes $L_\infty$ among all such equivalents (Xie and Li, 2024). $\square$

### D.5 Proof of Corollary 3.3

**Corollary D.5** (Restatement of Corollary 3.3). *Under the setup of Theorem 3.1, assume the training data satisfy the separability condition in Definition 3.1. If the DAE in (5) is under-parameterized with $p = \sum_{k=1}^{K} p_k \ll n$, then there exists a local minimizer of (6) of the form*

$$\boldsymbol{W}_2^\star = \boldsymbol{W}_1^\star = (\boldsymbol{W}_{\boldsymbol{X}_1} \quad \boldsymbol{W}_{\boldsymbol{X}_2} \quad \cdots \quad \boldsymbol{W}_{\boldsymbol{X}_K}) =: \boldsymbol{W}_{\mathrm{gen}},$$

*where each block $\boldsymbol{W}_{\boldsymbol{X}_k} \in \mathbb{R}^{d \times p_k}$ consists of the leading principal components of $\boldsymbol{X}_k \boldsymbol{X}_k^\top$ as in (8), and $\boldsymbol{W}_{\boldsymbol{X}_k} \boldsymbol{W}_{\boldsymbol{X}_k}^\top$ concentrates to the rank-$p_k$ optimal denoiser for $\mathcal{N}(\boldsymbol{\mu}_k, \boldsymbol{\Sigma}_k)$:*

$$\boldsymbol{W}_{\boldsymbol{X}_k} \boldsymbol{W}_{\boldsymbol{X}_k}^\top \;\to\; \left[(\boldsymbol{S}_k - \tfrac{\lambda}{\rho_k}\boldsymbol{I})(\boldsymbol{S}_k + \sigma^2\boldsymbol{I})^{-1}\right]_{\mathrm{rank}\text{-}p_k},$$

*where $\boldsymbol{S}_k$ is introduced in (10) and $\rho_k$ is the weight of the $k$-th mixture component. Moreover, when $\lambda \to 0$, the expected test loss (generalization error) satisfies*

$$\mathbb{E}_{\boldsymbol{X} \sim p_{gt}}[\mathcal{L}_{\boldsymbol{X}}(\boldsymbol{W}_2^\star, \boldsymbol{W}_1^\star)] \;\lesssim\; \sum_{k=1}^{K} \rho_k \left\{ \sum_{j \leq p_k} \frac{\mathrm{eig}_j(\boldsymbol{S}_k)\,\sigma^4}{\left(\mathrm{eig}_j(\boldsymbol{S}_k) + \sigma^2\right)^2} + \sum_{j > p_k} \mathrm{eig}_j(\boldsymbol{S}_k) + \frac{C_k\,p_k}{\sigma^2\,n_k} \right\},$$

*where $C_k > 0$ depends on $\sigma$ and spectral properties of $\boldsymbol{S}_k$, and $\mathrm{eig}_j(\boldsymbol{S}_k)$ denotes the $j$-th eigenvalue of $\boldsymbol{S}_k$ (independent of $d$).*

*Proof. Notation.* For a PSD matrix $A$, define $\boldsymbol{f}(\boldsymbol{A}) := (\boldsymbol{A} - \tfrac{\lambda}{\rho_k}\boldsymbol{I})(\boldsymbol{A} + \sigma^2\boldsymbol{I})^{-1}$, and let $\boldsymbol{f}_{p_k}(\boldsymbol{A})$ be $\boldsymbol{f}(\boldsymbol{A})$ truncated to its top $p_k$ eigendirections. Set

$$\delta_{p_k} := \mathrm{eig}_{p_k}(\boldsymbol{S}_k) - \mathrm{eig}_{p_k+1}(\boldsymbol{S}_k) > 0, \qquad r_{\mathrm{eff},k} := \mathrm{Tr}(\boldsymbol{S}_k)/\|\boldsymbol{S}_k\|_{\mathrm{op}}.$$

All high-probability statements are with respect to the draw of $\boldsymbol{X}_k$; $C > 0$ denotes a universal constant.

**(1) Plug in Theorem 3.1.** By Theorem 3.1, in a neighborhood of a block-structured point the DAE loss decouples across clusters, and each block solves a regularized LAE on $\boldsymbol{X}_k$ with effective noise weight $n_k\sigma^2$ and decay $n\lambda$. Hence the learned denoiser on cluster $k$ is $\widehat{D}_k := \boldsymbol{W}_{\boldsymbol{X}_k} \boldsymbol{W}_{\boldsymbol{X}_k}^\top$.

**(2) Concentration to the population denoiser.** For Gaussian clusters, $\frac{1}{n_k}\boldsymbol{X}_k\boldsymbol{X}_k^\top$ concentrates around $\boldsymbol{S}_k$. The LAE solution depends smoothly on its Gram matrix; combining this with a Davis-Kahan perturbation yields

$$\left\|\widehat{D}_k - \boldsymbol{f}_{p_k}(\boldsymbol{S}_k)\right\|_{\mathrm{F}} \;\leq\; \left(\tfrac{1}{\sigma^2} + \tfrac{C}{\delta_{p_k}}\right) \left\|\tfrac{1}{n_k}\boldsymbol{X}_k\boldsymbol{X}_k^\top - \boldsymbol{S}_k\right\|_{\mathrm{F}}.$$

Moreover, with probability at least $1 - e^{-t}$,

$$\left\|\tfrac{1}{n_k}\boldsymbol{X}_k\boldsymbol{X}_k^\top - \boldsymbol{S}_k\right\|_{\mathrm{F}} \;\lesssim\; \|\boldsymbol{S}_k\|_{\mathrm{op}} \sqrt{\tfrac{p_k\,(r_{\mathrm{eff},k}+t)}{n_k}}.$$

Combining the last two displays gives the explicit deviation

$$\left\|\widehat{D}_k - \boldsymbol{f}_{p_k}(\boldsymbol{S}_k)\right\|_{\mathrm{F}} \;\lesssim\; \|\boldsymbol{S}_k\|_{\mathrm{op}} \left(\tfrac{1}{\sigma^2} + \tfrac{1}{\delta_{p_k}}\right) \sqrt{\tfrac{p_k\,(r_{\mathrm{eff},k}+t)}{n_k}} \qquad \text{(w.h.p.).} \tag{19}$$

**(3) Population rank-$p_k$ DAE risk.** Let $\boldsymbol{x}' \sim \mathcal{N}(\boldsymbol{\mu}_k, \boldsymbol{\Sigma}_k)$ and $\boldsymbol{\varepsilon} \sim \mathcal{N}(\boldsymbol{0}, \boldsymbol{I})$ be independent. Define

$$\mathcal{L}_k^{\mathrm{pop}}(p_k) := \mathbb{E}\big[\|f_{p_k}(\boldsymbol{S}_k)(\boldsymbol{x}' + \sigma\boldsymbol{\varepsilon}) - \boldsymbol{x}'\|_2^2\big].$$

Diagonalizing $\boldsymbol{S}_k$ and using $f(A) = A(A + \sigma^2 I)^{-1}$ gives

$$\mathcal{L}_k^{\mathrm{pop}}(p_k) = \sum_{j \leq p_k} \frac{\mathrm{eig}_j(\boldsymbol{S}_k)\,\sigma^4}{\left(\mathrm{eig}_j(\boldsymbol{S}_k) + \sigma^2\right)^2} \;+\; \sum_{j > p_k} \mathrm{eig}_j(\boldsymbol{S}_k).$$

**(4) Generalization loss on cluster $k$.** Let $D_k^\star := f_{p_k}(\boldsymbol{S}_k)$. Then

$$\mathbb{E}\big[\|\widehat{D}_k(\boldsymbol{x}' + \sigma\boldsymbol{\varepsilon}) - \boldsymbol{x}'\|_2^2\big] = \mathcal{L}_k^{\mathrm{pop}}(p_k) + \mathrm{Tr}\big(\boldsymbol{S}_k\,(\widehat{D}_k - D_k^\star)^2\big) \leq \mathcal{L}_k^{\mathrm{pop}}(p_k) + \|\boldsymbol{S}_k\|_{\mathrm{op}}\,\|\widehat{D}_k - D_k^\star\|_{\mathrm{F}}^2.$$

Plug (19) into the last inequality to obtain, with probability at least $1 - e^{-t}$,

$$\mathbb{E}\big[\|\widehat{D}_k(\boldsymbol{x}' + \sigma\boldsymbol{\varepsilon}) - \boldsymbol{x}'\|_2^2\big] \leq \mathcal{L}_k^{\text{pop}}(p_k) + C \|\boldsymbol{S}_k\|_{\text{op}}^3 \Big(\tfrac{1}{\sigma^2} + \tfrac{1}{\delta_{p_k}}\Big)^2 \frac{p_k (r_{\text{eff},k} + t)}{n_k}. \quad (20)$$

This makes the $1/n_k$ rate and its dependence on $\sigma, \delta_{p_k}$ and the spectrum of $\boldsymbol{S}_k$ explicit.

**(5) From clusters to the mixture (population) bound.** Let $p_{\text{gt}} = \sum_{k=1}^K \rho_k \mathcal{N}(\boldsymbol{\mu}_k, \boldsymbol{\Sigma}_k)$. By linearity of expectation,

$$\mathbb{E}_{\boldsymbol{X} \sim p_{\text{gt}}}[\mathcal{L}_{\boldsymbol{X}}(\boldsymbol{W}_2^\star, \boldsymbol{W}_1^\star)] = \sum_{k=1}^K \rho_k \, \mathbb{E}\big[\|\widehat{D}_k(\boldsymbol{x}' + \sigma\boldsymbol{\varepsilon}) - \boldsymbol{x}'\|_2^2\big].$$

Apply (20) to each term and take a union bound over $k = 1, \ldots, K$ by choosing $t = \log(K/\eta)$. With probability at least $1 - \eta$,

$$\mathbb{E}_{\boldsymbol{X} \sim p_{\text{gt}}}[\mathcal{L}_{\boldsymbol{X}}(\boldsymbol{W}_2^\star, \boldsymbol{W}_1^\star)] \leq \sum_{k=1}^K \rho_k \left[\mathcal{L}_k^{\text{pop}}(p_k) + C \|\boldsymbol{S}_k\|_{\text{op}}^3 \Big(\tfrac{1}{\sigma^2} + \tfrac{1}{\delta_{p_k}}\Big)^2 \frac{p_k (r_{\text{eff},k} + \log(K/\eta))}{n_k}\right].$$

Absorbing $r_{\text{eff},k}$ and $\log(K/\eta)$ into a cluster-dependent constant $C_k$ yields exactly the last term in the corollary statement. (If one prefers a bound in *expectation* without failure probability, the same inequality holds with the right-hand side plus an $O(\eta)$ additive term by integrating the tail; choosing $\eta = n^{-2}$ makes this negligible.)

$\square$

## D.6    PROOF OF COROLLARY 3.4

**Corollary D.6** (Restatement of Corollary 3.4). *Let $X = [\boldsymbol{X}_1, \ldots, \boldsymbol{X}_K]$ satisfy Definition 3.1, where for $\ell = 1, \ldots, m$, $\boldsymbol{X}_\ell = (\boldsymbol{x}_\ell, \ldots, \boldsymbol{x}_\ell)$ is rank 1, and $\boldsymbol{X}_{m+1}, \ldots, \boldsymbol{X}_K$ contain distinct empirical samples from the remaining Gaussian modes. Suppose a ReLU DAE is trained with weight decay $\lambda \geq 0$ and input noise $\sigma > 0$. Then there exists a local minimizer of the form*

$$\boldsymbol{W}_2^\star = \boldsymbol{W}_1^\star = (r_1\boldsymbol{x}_1 \quad \cdots \quad r_m\boldsymbol{x}_m \quad \boldsymbol{W}_{\boldsymbol{X}_{m+1}} \quad \cdots \quad \boldsymbol{W}_{\boldsymbol{X}_K}),$$

*where the first $m$ columns memorize the duplicated clusters (as in Cor. 3.2), and the remaining blocks $\boldsymbol{W}_{\boldsymbol{X}_k}$ implement generalization on the nondegenerate clusters (as in Cor. 3.3).*

*Proof.* The proof follows by combining Cor. 3.2 and Cor. 3.3 and using the block-wise structure guaranteed by Thm. 3.1. In particular, Thm. 3.1 allows us to treat each cluster $\boldsymbol{X}_k$ independently at a local minimizer.

For the first $1 \leq j \leq m$ clusters, $\boldsymbol{X}_j$ is rank 1 and Cor. 3.2 implies that the corresponding columns of $\boldsymbol{W}_1^\star$ and $\boldsymbol{W}_2^\star$ are simply scaled data vectors $r_j\boldsymbol{x}_j$. For the remaining clusters $\boldsymbol{X}_{m+1}, \ldots, \boldsymbol{X}_K$, Cor. 3.3 yields the blocks $\boldsymbol{W}_{\boldsymbol{X}_k}$ that implement generalization on the nondegenerate modes. Stacking these columns and blocks gives precisely the stated form of $\boldsymbol{W}_1^\star = \boldsymbol{W}_2^\star$.

This corollary illustrates the local adaptivity of ReLU DAE models: they can memorize duplicated subsets while simultaneously generalizing on well-sampled regions of the data distribution. $\square$

