# OpenReview forum: "Generalization of Diffusion Models Arises with a Balanced Representation Space"
_ICLR.cc/2026/Conference — ICLR 2026 Poster_

### Official Review · Reviewer_6mi7 · 2025-10-28

**Soundness:** 3
**Presentation:** 3
**Contribution:** 1
**Rating:** 2
**Confidence:** 4

**Summary:**

In this work, authors study the representation space of diffusion models, showing especially for simplified scenario with 2-layered DAE that memorization can be attributed to the spikiness of activations. In particular, authors show that for overparameterized models examples are encoded in single neurons, while this is not the case for generalized underparametrized scenario.

**Strengths:**

- Insightful formulation with ReLU DAE which clearly introduces the main claims of the paper
- The theoretical contribution of the paper is clear and sound

**Weaknesses:**

- The paper's central observation—that memorization is linked to "spiky" neural activations—appears to be a rediscovery of the main finding from [1]. The authors propose using "the standard deviation of intermediate features" as a proxy for this spikiness. This metric is functionally identical to the z-score introduced in [1], which also identifies memorized examples by measuring the number of standard deviations an individual neuron's activation is from the mean.
- The analysis is restricted to a comparison between two well-understood extremes: an overparameterized model that memorizes and an underparameterized model that generalizes. This setup does not address the more complex and realistic scenario where a single, large-scale model simultaneously memorizes some training examples while generalizing others.
- In Section 3.1 there is an interesting example of memorization in overparametrized setup, where authors show that small DAE with large enough latent space (bigger than the dataset) can be optimized by memorizing individual samples (following block-wise structure). While this is a viable example, I am not sure that the conclusions are rigorous enough. I agree with comments presented in the first dot, but given the fact that the analysed solution is only the one of many local minimas, we cannot be sure that other minimas do not promote some generalization via neurons entanglement.
- The paper makes the strong claim that “our findings show that the representation space is not a byproduct but a determining factor for generation.” This implies a causal relationship that the experiments do not support.
- The conclusions on steering and editing representations are drawn from an insufficient sample size of just 8 qualitative examples. There is no quantitative evaluation to validate these claims.

**Questions:**

-

---

> ### Author Response · Authors · 2025-11-21
> **Response to Reviewer 6mi7 (1/2)**
>
> We thank the reviewer for the summary and for recognizing the theoretical contribution. We address the concerns point by point below.
>
> 1. **The difference between our findings and [1].**
>
>    We thank the reviewer for bringing this important reference to our attention. Although the exact citation was not listed in the review, to the best of our knowledge, we believe the reviewer is referring to:
>    *[1] Hintersdorf et al., Finding NeMo: Localizing Neurons Responsible For Memorization in Diffusion Models, NeurIPS 2024.*
>
>    If this is not the correct reference, we will gladly revise and cite the intended work. If [1] is indeed the reference, we agree that both works study feature-space signatures of memorization (spikiness vs. z-scores). However, there are substantial differences in focus and methodology. We now explicitly acknowledge [1] and carefully discuss these differences in Section 4.1 of the revised manuscript.
>
>    a) **Difference in research focus.**
>
>       As we understand it, the primary goal of [1] is to develop a practical memorization detection method, focusing on how text prompts induce memorization via the cross-attention module. In contrast, our paper focus on a theoretical explanation of generalization vs. memorization. By analyzing the optimal solution of a two-layer nonlinear ReLU-DAE, we aim to explain why spiky vs. balanced representations arise. For us, the detection of memorization through feature spikiness serves as a validation and a direct application of our theoretical findings, rather than a sole objective of our work.
>
>    b) **Differences in detection mechanism and scope.**
>
>       Moreover, our method applies to both conditional (T2I) and unconditional models without requiring the original text prompt, whereas “Finding NeMo” [1] focus on T2I models and focuses on prompt-triggered memorization via cross-attention layers. In contrast, we detect spiky representations in more general intermediate features, including feedforward layers distinct from cross-attention, and show that this spikiness persists even when the memorized image is presented without its original text prompt.
>
>    To make these points clear, we have added an empirical comparison with [1] in Table 1 of the revised paper and a dedicated discussion in Section 4.1 Line 460-462, carefully acknowledging and comparing these two approaches.
>
> 2. **The analysis is restricted to two well-understood extremes and does not address more complex and realistic scenarios.**
>
>    We thank the reviewer for bringing this important point to our attention. Real-world models indeed simultaneously memorize and generalize. In our framework, such models can be viewed as a set of local ReLU-DAEs that implement Cor. 3.2 or Cor. 3.3 in different input regions (areas around different training samples). Whether Cor. 3.2 or Cor. 3.3 holds in a given region determines whether the diffusion model memorizes or generalizes there.
>
>    Theoretically, our results do cover the more realistic setting where memorization and generalization coexist. We have added this study to the main text (Section 3.3, Corollary 3.4). In this hybrid case, we demonstrate that the model employs a mixed strategy based on sample complexity per Gaussian cluster. For clusters with few samples (rank-1 / singleton point), the corresponding weight columns tend to memorize and store individual samples. In contrast, for clusters with sufficient data, the corresponding columns capture local statistics. This yields hybrid column blocks in the learned weights that directly reflect the realistic situation where a single model both memorizes some examples and generalizes on others.
>
>    We also note that the theoretical understanding of generalization in diffusion models is still limited. Existing work often focuses on memorization via overparameterization [2] or on linear/idealized settings [3, 4]. In contrast, our analysis is carried out for a nonlinear two-layer DAE and explicitly characterizes the conditions under which the optimal weights encode statistics rather than individual samples. Together, these results provide a more realistic picture of how memorization and generalization can coexist in a single nonlinear model.

---

> ### Author Response · Authors · 2025-11-21
> **Response to Reviewer 6mi7 (2/2)**
>
> 3. **As the analysed solution is only the one of many local minimas, we cannot be sure that other minimas do not promote some generalization via neurons entanglement.**
>
>    We agree that the nonlinear objective contains many local minima. Our analysis focuses on a particular sparse, block-structured minimizer because, empirically, this is the solution that standard optimizers tend to find. As discussed in Corollary 3.2 and Figure 3 (left), we observe that practical adaptive optimizers such as Adam consistently converge to this sparse solution, which we attribute to algorithmic implicit bias [5, 6, 7].
>
>    To address the reviewer’s concern, we added new experiments in the revision (Figure 14 in Appendix C.2): in a highly overparameterized regime, we vary independent random seeds and optimizers (e.g., RMSProp, Adam, AdamW and Muon). In all four cases, the learned weights converge (up to permutation) to essentially the same sparse pattern. This supports our claim that implicit bias, particularly adaptivity, drives training toward the specific minimizer we analyze.
>
> 4. **The paper makes the strong claim that “our findings show that the representation space is not a byproduct but a determining factor for generation.” This implies a causal relationship that the experiments do not support.**
>
>    We thank the reviewer for this valuable suggestion. In the revision, we have downplayed our conclusion regarding a causal relationship in the Introduction and Conclusion from ‘determining’ to be “crucial and controllable”.
>
>    We do not intend to imply direct causality. Our reasoning is that generalization fundamentally arises from the neural network's successful approximation of an underlying (ground truth) denoiser. From this perspective, representation learning emerges as a highly interpretable and crucial aspect of the learning process. Our work thus aims to understand and describe this process primarily through the lens of learned representations.
>
> 5. **Limited number of steering/editing examples and lack of quantitative evaluation**
>
>    We agree that the original steering/editing experiments were limited in scope. In the revision, we have added steering experiments on SD 3.5 (Figure 20, Appendix C.4) and conducted additional ablation studies for SD 1.4 (Figures 18 and 19, Appendix C.4). Please also refer to Point 2 in our response to Reviewer Eg8y.
>
>    We also clarify in Section 4.2 that our goal is not to introduce a new state-of-the-art editing method or to quantitatively outperform existing steering techniques. Instead, the primary purpose of our steering experiments is to illustrate and deepen our understanding of the fundamental differences between the representation structures of memorized and generalized samples, thereby confirming the insights provided by our theoretical analysis. The study provides deep insights into the working mechanisms of the model in different regimes. A more extensive quantitative evaluation of editing performance is an interesting direction for future work.
>
> References
> [1] Hintersdorf et al., *Finding NeMo: Localizing Neurons Responsible For Memorization in Diffusion Models。* NeurIPS 2024.
> [2] Li, Sixu, Shi Chen, and Qin Li. *A good score does not lead to a good generative model.* arXiv preprint 2024
>  [3] George, Anand Jerry, Rodrigo Veiga, and Nicolas Macris. *Denoising score matching with random features: Insights on diffusion models from precise learning curves.* arXiv preprint 2025
>  [4] Li, Xiang, et al. *Understanding generalizability of diffusion models requires rethinking the hidden gaussian structure.* NeurIPS 2024
> [5] Xie, Shuo, Mohamad Amin Mohamadi, and Zhiyuan Li. *Adam Exploits $\ell_\infty $-geometry of Loss Landscape via Coordinate-wise Adaptivity.* ICLR2025
> [6] Xie, Shuo, and Zhiyuan Li. *Implicit Bias of AdamW: $\ell_∞ $-Norm Constrained Optimization.* ICML2024
> [7] Zhang, Yushun, et al. *Why Transformers Need Adam: a Hessian Perspective.* NeurIPS 2024

---

### Official Review · Reviewer_jzL9 · 2025-10-31

**Soundness:** 4
**Presentation:** 4
**Contribution:** 3
**Rating:** 8
**Confidence:** 4

**Summary:**

The authors present a study on diffusion model generalization. They use a two-layer Relu network in a denoising autoencoder framework. In this framework, and assuming a type of cluster-separability, the authors present a theorem relates the optimal weights to the data clusters. They then proceed to derive properties for the low and high data limits, demonstrating they relate to memoization and generalization, and explain how these two domains have different weight properties. The authors then demonstrate these properties exist in pretrained diffusion models and how they can explain limitations of "steering" where the data is sparse.

**Strengths:**

Overall I very much enjoyed reading this paper, here are some specific points of strength:
- The paper is clearly organized, and concepts are well explained and presented.
- I very much like the paper is centered around rigorous theory, but makes it approachable in presentation, and demonstrates the strengths and downstream applications their theory through practical experiments with real datasets and models.
- This is a paper that makes a strong contribution toward understanding diffusion model generalization fundamentals, which is still quite poorly understood. The authors not only deliver great insight for people working on understanding this generaliation, but also make it approachable for practitioners working on downstream tasks

**Weaknesses:**

- There are a few things about definition 3.1 that seem odd to me. What 3.1 says is that the cluster means must be separated by at least some angle related to beta. This seems like a very specific type of clustering. It excludes for example clustering in the same direction (imagine a multi-modal gaussian with a sequence of modes in one dimension). Perhaps this is a common assumption I am not aware of. Could the authors defend this choice? Is it necessary for the theorems?
- It seems to me that depending on beta, the assumption the authors use sets some limitations on the ratio between the number of clusters and the data dimension. When I think about such clusters and data dimension in the image space, I would argue that the number of clusters is much larger than the data dimension. This would limit beta for such datasets, and because of that make theorem 3.1 less applicable since it assumes beta < 1. Can the authors comment?
- A small comment here is that the "margin" gamma (line 184) does not seem to be defined in the text.
- It seems surprising to me that the "spikiness" observation from Cor. 3.2 holds in more complex networks than two-layer Relu networks. While it seems like a perfectly reasonalbe intermediate state for the two-layer net, for more complex networks it seems like that would be much more complex. Beyond empirical results, can the authors give some intuition for why this would hold?

**Questions:**

- (see also weaknesses)
- Do the authors use sigma in the denoiser network? It is common practice for sigma to be part of the denoising representation. Would adding versus omitting it change any results?
- Can the authors confirm that in the low data case, the network's output matches the optimal empirical denoiser? It seems like this would be the case, but I did not find that back in the paper. It would be useful to the reader to make a statement about that.
- Related to the last question: is the reason that the under-parametrized networks do not output the optimal empirical denoiser that the network does not have enough capacity to learn it?

---

> ### Author Response · Authors · 2025-11-21
> **Response to Reviewer jzL9**
>
> We thank the reviewer for the clear summary of our work and for recognizing both the theoretical and empirical contributions. We also appreciate the thoughtful questions, which we address point by point below.
>
> 1. **Definition 3.1 and separability assumptions.**
>
>    Thanks for pointing this out. Definition 3.1 (and requiring $\beta < 0$) is indeed a non-trivial separability assumption and can be relaxed. Our choice is primarily driven by the need to separate clusters using bias-free linear layers, so we impose separation based on angle/direction. This makes the theorems simpler and more accessible by the reader. With more dedicated analysis, we believe that we can relax the assumption to standard hyperplane separability [1] and show a similar result by using a ReLU-DAE with bias. In short, the key phenomena we identified (i.e., learning raw data vs. statistics and spiky vs. balanced representations) do not rely critically on this exact separability condition. As we discuss in Appendix C.2 (Lines 916–956), the same behavior persists even under weaker separability assumptions.
>
> 2. **Margin $\gamma$ (Line 184)**
>
>    We thank the reviewer for the catch! The margin $\gamma$ is meant to quantify how negatively a sample from one cluster can align with samples from other clusters. We have added a more detailed description of $\gamma$ to the main text, including its calculation method and the location of its full form, in a footnote following its first occurrence (Page 4).
>
> 3. **Spikiness beyond two-layer ReLU networks.**
>
>    In a localized view, the learning behavior of practical models remains similar to that of a ReLU-DAE we studied theoretically, and the spiky vs. balanced representation regimes in Corollary 3.2 and Corollary 3.3 can be applied locally. The spikiness of the representations indicates which regime is in effect. Around a given input, a deep network behaves approximately linearly [2,3], so each local region can be viewed as implementing a mechanism close to one of our two regimes (see also the **Overall Response** for more discussion).
>
>    More intuitively, if a model memorizes some data, it must internally store a template for that data within a small set of neurons. When it encounters this data again, those neurons react strongly, producing large activations at a few coordinates and resulting in a spiky representation. In contrast, when the model encodes shared statistics rather than specific samples, many units contribute, resulting in a more balanced representation.
>
> 4. **Effect of including $\sigma$ (noise conditioning) in the ReLU-DAE**
>
>    In our theory, we currently consider training the denoiser network at a single fixed $\sigma$ and do not explicitly include it as an input to the model. This choice simplifies the analysis. In principle, the results can be extended by adding a time/noise conditioning based on $\sigma$ and applying the same proof ideas across multiple noise levels. Moreover, prior work (e.g., [4]) shows that representations often exhibit a form of consistency across different $\sigma$ values, and we hypothesize that this is related to the consistency of the learning outcome as described after Corollary 3.3 (Line 397–401).
>
> 5. **Connection to the optimal empirical denoiser**
>
>    In Point 2 of the remarks for Corollary 3.2 (Case 1), we are precisely trying to argue that the learned network approaches the theoretical empirical optimum, in the sense that both map a noisy input toward (roughly) its nearest training image and achieve a dimension-independent low empirical loss similar to the optimum. As a result, our solution in Case 1 behaves like a memorizing denoiser. We now make the definition of the empirical denoiser and its closeness to our solution explicit in the preliminaries (Line 114–130) and in the remark (Line 263–269).
>
>    This approximation is indeed partially driven by model capacity: in the under-parameterized regime, the network lacks sufficient degrees of freedom to represent the full empirical denoiser, and instead learns a smoother approximation that captures local statistics and generalizes [5,6]. However, this is not the only factor; the learning dynamics [5] and loss landscape [6] also play crucial roles for enabling generalization.
>
> References:
> [1] Duda, Richard O., and Peter E. Hart. *Pattern Classification*. John Wiley & Sons, 2006.
> [2] Wang, Binxu. *An Analytical Theory of Power Law Spectral Bias in the Learning Dynamics of Diffusion Models.* NeurIPS 2025.
> [3] Lukoianov, Artem, et al. *Locality in Image Diffusion Models Emerges from Data Statistics.* NeurIPS 2025.
> [4] Li, Xiao, et al. *Understanding Representation Dynamics of Diffusion Models via Low-Dimensional Modeling.* NeurIPS 2025.
> [5] Bonnaire, Tony, et al. *Why Diffusion Models Don’t Memorize: The Role of Implicit Dynamical Regularization in Training.* NeurIPS 2025.
> [6] Chen, Zhengdao. *On the Interpolation Effect of Score Smoothing.* arXiv preprint, 2025.

---

### Official Review · Reviewer_NqY1 · 2025-10-31

**Soundness:** 3
**Presentation:** 3
**Contribution:** 3
**Rating:** 6
**Confidence:** 3

**Summary:**

This paper aims to understand memorisation and generalisation of diffusion models from the representation perspective. Specifically, the authors built a theoretical framework in a two-layer ReLU diffusion model, which is more manageable in the theoretical sense. Firstly, the authors analysed the optimal solution of the above simplified diffusion model, which is over-parameterized. Through the analysis, they found that spiky representations happened, which are a a signal of memorisation. Secondly, the authors analysed the scenario of under-parameterisation, and showed that balanced representation as a signature of generalisation. Finally, the authors demonstrate the theoretical analysis can result in real-world impact, such as better memorisation detection metric and a steering approach for image editing.

**Strengths:**

This paper benefits from both theoretical analysis and further empirical impact. Especially, through the theoretical analysis, the authors showed that data representations (spikiness or balance) could be signals for memorisation/generalisation. Such signals could be used for memorisation detection, and have high accuracy than previous metrics and meanwhile is prompt-free.

**Weaknesses:**

I have the following concerns or questions, which may need authors' clarifications. Also please correct me if  I am wrong.

1. In case 1 of over-parameterisation, the authors found that with the optimal solution, the representations of a **single training sample** exhibits spikiness (in line 269). I am wondering if we input a different sample (not in training data) to such a neural network, whether the learned representations become a zero vector.

2. In case 2 of under-parameterisation, the authors found that with the optimal solutions, the representations of a **single training sample** exhibits balance (in line 369), hence smaller std. I am also wondering what would happen if we input a different sample (not in training data, or a generalised sample).

3. It seems that the authors consider two neural networks, one could memorise all samples, one could generalise new samples. However, for a trained diffusion model, it could both memorise and generalise. Why is the spikiness in representations an indicator for memorisation?

4. The authors mainly discuss about optimal solutions for diffusion model. However, [1,2,3] show that as long as the number of training samples is finite, whether diffusion model is over-parameterised or not, there exists a theoretical optimum which could always generate memorised training data. Can you clarify the connection between such a theoretical optimum and the optimal solution shown in this paper? Is this because the model family of two-layer ReLU network cannot represent the theoretical optimum?

References:\
[1] Yi et al. On the generalization of diffusion model. 2023.\
[2] Gu et al. On memorization in diffusion models. 2023.\
[3] Kamb et al. An analytic theory of creativity in convolutional diffusion models. ICML 2025.

**Questions:**

See the weakness.

---

> ### Author Response · Authors · 2025-11-21
> **Response to Reviewer NqY1**
>
> We thank the reviewer for the clear summary of our work and for recognizing both the theoretical and empirical contributions. We also appreciate the thoughtful questions, which we address point by point below.
>
> 1. **Representations of unseen samples for ReLU-DAE in memorization regime.**
>
>    We thank the reviewer for bringing up this point. In Case 1, the representation of an unseen (test) sample is determined by its angle with the memorized training images (which form the learned data matrix). If the test sample has a large (obtuse) angle with all training images, its ReLU activations vanish, and we obtain a zero representation vector as the reviewer pointed out. If it has a positive correlation with some training images, those coordinates become active and we obtain a representation that is still sparse, but typically less spiky than that of the memorized training samples.
>    To verify this behavior empirically, we have added new results on the representations of test CelebA samples under our setting in Figure 17 of Appendix C.3
>
> 2. **Representations of unseen/generalized samples for ReLU-DAE in the generalization regime.**
>
>    In Case 2, the representation of an unseen input depends on how it aligns with the learned data statistics. If the test sample comes from the same distribution as the training set, it can be encoded by the learned statistics, and its representation will be just as balanced as those of the training samples, again reflecting “generalization” in our framework. If the test sample is strongly out-of-distribution (e.g., nearly orthogonal to all training samples), its representation will again be close to zero (please also see Figure 17 in Appendix C.3).
>
> 3. **Why spikiness indicates memorization when real diffusion models both memorize and generalize.**
>
>    In real diffusion models, memorization and generalization do co-exist [4, 5]: the model behaves differently in different regions of the input space. Our formulation is therefore best viewed as a localized analysis of real models: in each local region, the behavior is well-approximated by one of our two regimes. In this view, spiky representations indicate that the local behavior is close to Corollary 3.2 (memorization), whereas balanced representations indicate that Corollary 3.3 (generalization) is happening in that region.
>    Moreover, we also prove that this coexistence in our ReLU-DAE setting (see Corollary 3.4 of the revised paper) where the result demonstrates an interpolation between the two regimes of memorization and generalization. Please also refer the reviewer to our **Overall Response** for a more detailed discussion of this local view.
>
> 4. **Connection to the theoretical ERM optimum (“empirical denoiser”) in [1,2,3].**
>
>    [1,2,3] show the existence of an ERM optimum that perfectly fits the finite training set, so there is always a theoretical optimum that can generate memorized training data. In practice, however, we train a parameterized network using gradient-based optimization. The fact that we typically do not reach this ERM optimum (except in extremely small-data cases) is due to the inductive biases and smoothing effects introduced by the parameterization and training dynamics [3, 6, 7], which we study explicitly using a two-layer ReLU model.
>    As noted in Point 2 of the remark following Corollary 3.2 (Page 5, Line 263-269), our learned network can be seen as a “raw” approximation of the ERM optimum: both roughly map each input toward its nearest training sample and achieve very low empirical loss, thereby reproducing and memorizing training samples. In Corollary 3.3, we similarly argue that the learned solution mimics the ground-truth denoiser for the underlying distribution, leading to generalization (Page 7, Line 360-368). We have made this connection more explicit in the preliminaries of our revised paper (see Line 114-130).
>
> References:
>  [1] Yi et al. *On the Generalization of Diffusion Model.* arXiv preprint, 2023.
>  [2] Gu et al. *On Memorization in Diffusion Models.* TMLR.
>  [3] Kamb et al. *An Analytic Theory of Creativity in Convolutional Diffusion Models.* ICML 2025.
>  [4] Ross, Brendan Leigh, et al. *A Geometric Framework for Understanding Memorization in Generative Models.* ICLR 2025.
>  [5] Somepalli, Gowthami, et al. *Understanding and Mitigating Copying in Diffusion Models.* NeurIPS 2023.
>  [6] Niedoba, Matthew, et al. *Towards a Mechanistic Explanation of Diffusion Model Generalization.* ICML 2025.
>  [7] Chen, Zhengdao. *On the Interpolation Effect of Score Smoothing.* arXiv preprint, 2025.

---

### Official Review · Reviewer_Eg8y · 2025-11-01

**Soundness:** 3
**Presentation:** 3
**Contribution:** 2
**Rating:** 6
**Confidence:** 3

**Summary:**

This paper investigates the mechanisms of memorization and generalization in diffusion models, starting with a theoretical analysis based on a two-layer ReLU denoising autoencoder.

The claim is that mem. or gen. is determined by the learned representations: (1) memorization corresponds to learning spiky, sample-specific representations and weights that store the training data sparsely, when model parameters $p$ are larger than data size $n$ ($p\geq n$), and (2) generalization arises when the model learns balanced representations that capture local data statistics, when it is under-parameterized ($p \ll n$).

Based on the analysis, the authors introduce two practical applications: a highly efficient and effective method for detecting memorized content by measuring the spikiness of representations, and a training-free image editing technique based on representation steering, which demonstrates that generalized samples are more editable than memorized ones. The paper also validates these methods on some real-world models, including EDM, DiT and SD1.4.

**Strengths:**

- The paper presents a valuable contribution by proposing a more fundamental, representation-centric framework to explain the behaviors of memorization and generalization in diffusion models.
- The quality of the theoretical analysis is high, and the findings are shown to be consistent with observations in real-world models, as validated through two practical applications.
- The memorization detection method demonstrates high performance and broad applicability across different models and datasets.
- The observation that generalized samples are more steerable, while memorized samples exhibit brittle editing behavior, provides a novel and interesting insight.

**Weaknesses:**

- The theoretical framework is built upon a two-layer DAE. While empirical results suggest the conclusions hold more broadly, I do not think it is very clear why the findings based on the linear projection dimension $p$ vs. data size $n$ ($p\geq n$ or $p \ll n$) should transfer to deep, multi-layered UNet / DiT.
- The image editing experiments are limited in scope and somewhat unclear in their details.
  - The representation steering method is only demonstrated on Stable Diffusion 1.4, which is somewhat old in this area. Its effectiveness and the observed behavior have not been tested on more recent models like DiT / MM-DiT based text-to-image models.
  - The paper lacks a justification for how the "encoder" $g_\theta$ and "decoder" $h_\theta$ are determined (Line 461, Lines 990-992). The appendix specifies that features are extracted from 6 distinct layers in up_blocks.0 and up_blocks.1, but how these specific layers are selected is not provided. Additionally, I assume they form a collection of features with the size $100\times C\times H\times W \times 6$. Are these features averaged, or is the steering performed on 6 layers in parallel?
- The image editing approach requires generating 100 reference images to compute a mean representation for the target concept, which makes it inapplicable to image-guided editing scenarios using a few (<10) provided reference images. Furthermore, will the reference image generation procedure affect the editing quality? For example, how the editing process would be affected if the generated reference samples were themselves memorized or of low quality.

**Questions:**

Please refer to the weaknesses regarding the image editing approach and experiments.

---

> ### Author Response · Authors · 2025-11-21
> **Response to Reviewer Eg8y**
>
> We thank the reviewer for their positive assessment of our theoretical contributions and for finding our memorization detection method valuable. We appreciate the constructive feedback regarding the scope of our experiments and technical clarifications. We address these points below:
>
> 1. **Transferability of 2-layer DAE theory to deep U-Net/DiT models: why findings based on projection dimension ($k$) vs. data size ($n$) in a simple model should transfer to deep networks.**
>
>     The core transfer is not about exact parameter–data size correlations, but about the mechanism of (representation) learning. Our theory isolates two distinct regimes: learning empirical data points (memorization & spiky representations) versus learning underlying statistics (generalization & balanced representations), which we also observe in practical deep models. As detailed in our **Overall Response**, deep diffusion models (U-Net or DiT) exhibit approximate piecewise linearity [1, 2]. Consequently, they can be viewed as a collection of local ReLU-DAEs. Whether the model memorizes or generalizes in a specific region then depends on the local sample complexity [3].
>
> 2. **Scope of editing:** the reviewer asks for experiments on more advanced T2I models, as well as more experimental details and justifications.
>
>     a)  **Testing on recent DiT-based models (SD 3.5).** We initially selected SD 1.4 because it is one of the few models with a publicly verified registry of memorized prompts, allowing for a controlled comparison between generalized and memorized behaviors. To address the reviewer’s suggestion, we applied our representation steering to Stable Diffusion 3.5, a recent DiT-based model. The results (Appendix C.5, Figure 20) demonstrate that our steering approach remains effective on this modern architecture. As there is no verified dataset of memorized prompts for SD 3.5 available yet, this evaluation is limited to generalized samples, but it confirms the method’s broader applicability beyond U-Nets.
>
>     b) **Clarification on “encoder/decoder” definitions and layer selection.** We appreciate the careful reading of our paper. Following recent works [4,5,6,7]  on studying representation learning in diffusion models , we adopt the convention of treating the first half of the network as the “encoder” and the second half as the “decoder.” We extract features from up_blocks.0 and up_blocks.1, as they are closest to the bottleneck layer and contain the most compact semantic information.
>
>    c) **Ablation study of layers.** To verify robustness, we added an ablation study (Appendix C.5, Figure 18) examining steering performance across different layer counts. We find that the method is robust; even using a single intermediate layer is sufficient to achieve effective steering and differentiate between generalized and memorized behaviors.
>
>    d) **Implementation detail.** For steering, the features are averaged across layers. And steering is performed on six selected layers in parallel. For each layer, we extract the feature tensor ($100 \times C \times H \times W$) and average across one hundred samples to obtain a per-layer steering vector ($1 \times C \times 1 \times 1$). This vector is then applied to that specific layer. We have revised Appendix D.3 to make these points clear
>
> 3. **Applicability to few-shot editing (<10 images) and reference quality.**
>
>    a) **Few-shot ablation.** We conducted a new ablation study (Appendix C.5, Figure 19) progressively reducing the number of reference images. We find the method remains robust with as few as 5–10 reference images, achieving comparable editing quality to the 100-image baseline.
>
>    b) **Reference quality.** The reviewer correctly notes that memorized or low-quality references could degrade steering. In principle, if reference images are memorized, their representations would be “spiky” (per our theory) and less effective for identifying a semantic direction. However, in our experiments, we explicitly avoid known memorized prompts for generating the reference images. This ensures the steering vector captures a generalized concept rather than a memorized one.
>
> References:
>  [1] Lukoianov, Artem, et al., *Locality in Image Diffusion Models Emerges from Data Statistics.* NeurIPS 2025
>  [2] Li, Xiang, et al., *Understanding Generalizability of Diffusion Models Requires Rethinking the Hidden Gaussian Structure.* NeurIPS 2024
> [3] Ross, Brendan Leigh, et al. *A geometric framework for understanding memorization in generative models.* ICLR 2025
>  [4] Xiang et al., *Denoising Diffusion Autoencoders are Unified Self-supervised Learners.* ICCV 2023
>  [5] Baranchuk et al., *Label-Efficient Semantic Segmentation with Diffusion Models.* ICLR 2022
>  [6] Kwon et al., *Diffusion Models already have a Semantic Latent Space.* ICLR 2023
>  [7] Chen, Xinlei, et al., *Deconstructing Denoising Diffusion Models for Self-Supervised Learning.* ICLR 2025

---

> > ### Comment · Reviewer_Eg8y · 2025-11-25
> >
> > I thank the authors for their detailed response and the effort put into the new experiments on image editing. Most of my previous concerns regarding SD 1.4 and the implementation details have been addressed. However, I am still not fully convinced about the contributions and significance.
> >
> > While the theoretical analysis and the framework appears solid, I personally think the conclusions are somewhat expected and not that "surprising", paricularly regarding the "over-parameterized vs. under-parameterized models" and "spiky vs. balanced representations". Furthermore, the practical implications seem limited:
> > - The image editing experiments remain largely qualitative, and appear similar to long-standing latent manipulation techniques (e.g., in GANs or earlier diffusion works). The new SD 3.5 experiements only show relatively simple manipulations (i.e., "+dark", "+red").
> > - It remains unclear how the "spiky vs. balanced" insight can actually guide us to improve diffusion models. For instance, if a model lacks generalization, would actively regularizing for more balanced/non-spiky representations during training lead to a improved model? I am currently not very convinced about that causality.

---

> ### Author Response · Authors · 2025-11-29
> **Response to Reviewer Eg8y (1/2)**
>
> We thank the reviewer for their timely response and constructive feedback, and for specifically emphasizing the importance of practical impact in this domain. We address the remaining concerns below by clarifying our contributions and providing additional experiments.
>
> ### 1. Theoretical and Practical Significance
>
> Understanding memorization and generalization in diffusion models is a fundamental and challenging problem. While recent studies have garnered significant attention (e.g., [1, 2, 3, 4, 5]), these findings often largely rely on empirical observations [1, 2] or focus on simplified surrogates (e.g., linear models [5] or random features [4, 6]), and therefore do not fully explain the phenomena observed in real-world diffusion models.
>
> In contrast, our work analyzes a nonlinear DAE and provides a complete characterization of both memorization and generalization. Crucially, our framework captures the **hybrid regime** in which memorization and generalization coexist (see Section 3.3 of our revised paper)—a practical setting [7, 8] that cannot be analyzed using prior frameworks. To the best of our knowledge, this is the first theoretical work to explicitly characterize how internal structure (i.e., learned representations) promotes generalization in diffusion models.
>
> ### 2. Practical Implications (New Experiments)
>
> Our theoretical findings are validated on real-world models and have enabled the following impactful applications:
>
> - **Memorization detection.** Our analysis of internal representation structures led us to propose a SoTA memorization detection method. Our approach is unique in being **prompt-free** and **highly efficient** for both T2I and unconditional diffusion models—a setting where prior methods [9, 10] are largely ineffective.
>
> - **(New Experiment) Preventing memorization via representation regularization.** Following the reviewer’s suggestion, we investigated using representation regularization to prevent memorization when training on limited data. We trained a DDPM on a 4,096-sample CIFAR-10 subset, adding an $\ell_2$ penalty on the bottleneck layer to suppress representation "spikiness."
>
>   As shown in Table 1, the baseline model suffers from overfitting and memorization [2] (indicated by increasing Test Loss and FID after Epoch 400). In contrast, the regularized model remains stable and continues to improve. We will incorporate these results into the revised paper (see Appendix F).
>
> **Table 1. Comparison of Regularized Training vs. Baseline on CIFAR-10 Subset (4,096 samples). Lower is better ($\downarrow$).**
>
> | **Epoch** | **Baseline FID (↓)** | **Baseline Test Loss (↓)** | **Reg. FID (↓)** | **Reg. Test Loss (↓)** |
> | --------- | -------------------- | -------------------------- | ---------------- | ---------------------- |
> | 100       | 51.86                | 102.08                     | 65.93            | 102.94                 |
> | 200       | 35.68                | 99.74                      | 43.57            | 100.16                 |
> | 300       | 28.69                | 99.90                      | 36.41            | 98.41                  |
> | 400       | **27.33**            | 106.97                     | 30.48            | 96.81                  |
> | 500       | 33.15                | 120.83                     | 26.26            | 96.47                  |
> | 600       | 42.46                | 138.62                     | 23.37            | 96.85                  |
> | 700       | 53.16                | 156.35                     | **22.09**        | 97.85                  |
> | 800       | 61.68                | 172.30                     | 22.39            | 99.07                  |
>
> This experiment empirically validates that constraining the representation space mitigates overfitting. It demonstrates that guiding the model toward balanced and informative representations improves generation quality, aligning with recent methods that enhance diffusion models via representation-space modifications.
>
> ### 3. On the Steering-Based Editing Method
>
> Finally, regarding the steering-based editing method, we acknowledge that our approach is simple and our evaluation is not exhaustive. However, our primary goal was not to propose a new state-of-the-art editing technique, but rather to **probe the fundamental differences** in representation between generalized and memorized samples.
>
> Our key finding (which the reviewer also noted as insightful) is that generalized representations preserve semantic information, enabling efficient and interpretable editing, whereas memorization actively impedes it. This experiment directly corroborates our core theoretical claims regarding structural differences in representations and provides practical guidance for effectively steering these methods.

---

> ### Author Response · Authors · 2025-11-29
> **Response to Reviewer Eg8y (2/2)**
>
> **References**
>
> [1] Kadkhodaie, Zahra, et al. *Generalization in diffusion models arises from geometry-adaptive harmonic representations.* ICLR 2024 (outstanding paper).
>
> [2] Zhang, Huijie, et al. *The emergence of reproducibility and consistency in diffusion models.* ICML 2024.
>
> [3] Kamb, Mason, and Surya Ganguli. *An analytic theory of creativity in convolutional diffusion models.* ICML 2025 (oral).
>
> [4] Bonnaire, Tony, et al. *Why diffusion models don't memorize: The role of implicit dynamical regularization in training.* NeurIPS 2025 (best paper).
>
> [5] Li, Xiang, et al. *Understanding generalizability of diffusion models requires rethinking the hidden Gaussian structure.* NeurIPS 2024.
>
> [6] Li, Puheng, et al. *On the generalization properties of diffusion models.* NeurIPS 2023.
>
> [7] Somepalli, Gowthami, et al. *Understanding and mitigating copying in diffusion models.* NeurIPS 2023.
>
> [8] Ross, Brendan Leigh, et al. *A geometric framework for understanding memorization in generative models.* ICLR 2025 (spotlight).
>
> [9] Wen, Yuxin, et al. *Detecting, explaining, and mitigating memorization in diffusion models.* ICLR 2024 (oral).
>
> [10] Hintersdorf, Dominik, et al. *Finding NeMo: Localizing neurons responsible for memorization in diffusion models.* NeurIPS 2024.

---

### Author Response · Authors · 2025-11-21
**Overall Response**

We thank all reviewers for their thoughtful and constructive feedback and are encouraged by the positive assessments of our work. In particular, the reviewers highlighted the quality of our theoretical analysis (all reviewers), the formulation of the ReLU-DAE model (6mi7), the strong connection between our theory and the practical behavior of diffusion models (jzL9, Eg8y), and the effectiveness of our two practical methods for memorization detection and image editing (NqY1, jzL9, Eg8y). During the rebuttal period, we have carefully addressed all questions raised by the reviewers and made corresponding revisions and additions to the manuscript; all changes are highlighted in blue. A recurring theme in the reviews concerns the relationship between our ReLU-DAE analysis and real diffusion models (EDM, DiT, SD1.4). We provide a clarification below:

1. **Role of the simplified model.** Many existing works analyze the generalization of diffusion models under strong simplifications, such as random feature models [1, 2] or fully linear models [3], which provide valuable but more idealized insights. In contrast, we study a more complex nonlinear two-layer ReLU-DAE model that provides a unified framework characterizing both memorization and generalization across different ratios of model capacity to training data size. Moreover, our results reveal distinct structures in the representation space (spiky versus balanced) in these regimes, which are consistently observed in practical diffusion models and thus support the intuition developed by our theoretical analysis.

2. **Connection to real diffusion models via local linearization.** Modern diffusion networks are approximately piecewise linear [3, 4, 5, 6], so their behavior around a given input can locally be captured by a piecewise-linear model (ReLU-DAE). This means real models can be considered as a collection of local ReLU-DAEs. In our framework, the spikiness of representations on practical diffusion models confirms this claim, that either Cor. 3.2 (memorization, spiky representation) or Cor. 3.3 (generalization, balanced representation) holds locally, and thus whether the real model memorizes or generalizes locally.

   Specifically, we compute the Jacobians of the denoisers in EDM and SD1.4 and analyze their SVDs (see Appendix C.1). Around generalized samples, the Jacobian exhibits slowly decaying singular values, and its singular vectors align with data statistics in practice (as predicted by Cor. 3.3). In contrast, around memorized samples, the Jacobian becomes very low-rank: the dominant singular vector corresponds closely to training samples, confirming that the model is effectively storing and denoising along the direction of specific training examples in practice (consistent with our theoretical findings in Cor. 3.2).

3. **Coexistence of memorization and generalization.** Our theory can capture the co-existence that happens in real-world diffusion models. To demonstrate this, we have added Corollary 3.4 Page 8 of the main text in the revised paper. This corollary shows that a single ReLU-DAE already captures the co-existence phenomenon: the optimal weight memorizes training samples from clusters with limited data while learning the underlying statistics for clusters with sufficient data. This is consistent with what we observe in large-scale diffusion models, where memorization and novel generation coexist in different regions of the image space [7, 8].

We hope these clarifications make the conceptual bridge between our theoretical framework and practical diffusion models more explicit and address the reviewers’ main concerns. In the following, we address each reviewer’s other questions one by one.

References:
[1] Bonnaire, Tony, et al. *Why Diffusion Models Don't Memorize: The Role of Implicit Dynamical Regularization in Training.* NeurIPS 2025
[2] George, Anand Jerry, Rodrigo Veiga, and Nicolas Macris. *Denoising score matching with random features: Insights on diffusion models from precise learning curves.* arXiv preprint 2025
[3] Li, Xiang, et al. *Understanding generalizability of diffusion models requires rethinking the hidden gaussian structure.* NeurIPS 2024
[4] Lukoianov, Artem, et al. *Locality in image diffusion models emerges from data statistics.* NeurIPS 2025
[5] Wang, Binxu. *An analytical theory of power law spectral bias in the learning dynamics of diffusion models.* NeurIPS 2025
[6] Wang, Binxu, and John J. Vastola. *The unreasonable effectiveness of gaussian score approximation for diffusion models and its applications.* TMLR
[7] Somepalli, Gowthami, et al. *Understanding and mitigating copying in diffusion models.* NeurIPS 2023
[8] Ross, Brendan Leigh, et al. *A geometric framework for understanding memorization in generative models.* ICLR 2025

---

### Author Response · Authors · 2025-12-04
**Summary of Contribution and Rebuttal Updates**

Dear Area Chair, We acknowledge the specific circumstances regarding the review process this year. To assist in your assessment, we provide a brief summary of our work and the rebuttal updates below.

## Contribution of the Work:

We prove two learning regimes for diffusion models under a nonlinear ReLU DAE framework: they either **compose data statistics to generalize** (producing balanced representations) or **store discrete data points to memorize** (producing spiky representations). This distinction is prevalent in real-world models, enabling us to achieve memorization detection, controllable editing, and enhanced generalization. In particular, our detection method achieves SoTA performance. We believe we are the first to theoretically investigate how internal representation structures enable generalization in diffusion models, establishing representations as a crucial and controllable factor for generation.

## Rebuttal:

All reviewers praised the insightfulness and novelty of our nonlinear ReLU-DAE framework, as well as the strong connection between our theory and practical behaviors (NqY1, jzL9, Eg8y). Based on their feedback, we made the following revisions:

1. **Representation-Output Connection (Eg8y & 6mi7):** We addressed inquiries regarding how representations affect output by demonstrating that representations from generalized samples are more effective for editing. We also conducted new experiments (**Appendix F**) showing that regularizing representations during training effectively prevents memorization and enhances generalization.
2. **Addressing Reviewer 6mi7:** We addressed 6mi7's concern regarding similarity to prior work. As the specific citation was omitted in the review, we identified *Finding NeMo* [1] as the likely intended reference and clarified the fundamental differences in **Section 4.1 and Table 1**. Since the reviewer did not respond to our rebuttal or complete the review, we kindly ask you to take our detailed response into consideration.
3. **Connection to Real-World Models:** We included **Corollary 3.4**, which captures the "hybrid" regime where real-world models memorize in areas with limited data but generalize in regions with sufficient data. We also provided justifications for viewing real-world models as a collection of local ReLU-DAEs.
4. **Clarifications for NqY1 and jzL9:** We made the theory more explicit by highlighting the connection between our learned solution, the empirical denoiser (Memorization), and the underlying denoiser (Generalization) on **page 3**.
5. **Technical Details:** We resolved ambiguities regarding steering details for Eg8y (**Appendix C.4**), representations of test samples for NqY1 (**Appendix C.3**), and the margin notation in the main Theorem for jzL9.

[1] Hintersdorf et al., Finding NeMo: Localizing Neurons Responsible For Memorization in Diffusion Models. NeurIPS 2024.

---

### Meta-Review · Area_Chair_zAcg · 2026-01-06

**Summary:**

The key decision factors were (i) whether the paper provides a novel and theoretically grounded understanding of memorization in diffusion models through internal representations, (ii) whether the connection to practical, real-world diffusion models is convincingly established and yields actionable tools (e.g., memorization detection, representation steering), and (iii) whether the proposed methods are original and practically impactful relative to prior work. Reviewers generally appreciated the theoretical framing and the representation-centric perspective, and several found the empirical connections to real models compelling. During rebuttal, the authors added experiments on newer models (including SD 3.5), expanded ablations, and introduced a representation-regularization strategy aimed at preventing memorization, strengthening the bridge between theory and practice. The remaining disagreement is largely about perceived novelty and the practical significance of the proposed tools, with one reviewer still viewing the contributions as incremental relative to prior memorization-localization work and questioning the strength of the evidence for broader impact.

**Reviewer Concerns:**

Addressed concerns: The rebuttal strengthened the empirical connection to modern diffusion models by adding experiments on SD 3.5 and providing additional ablations that clarify implementation choices and robustness (e.g., layer selection, number of reference images, and steering behavior). The authors also clarified how the simplified ReLU-DAE analysis can be interpreted in a localized, piecewise-linear view of deep diffusion networks, and added a hybrid-regime characterization to reflect the realistic coexistence of memorization and generalization within a single model. In response to requests for actionable implications, the rebuttal included a regularization-based training strategy that empirically suppresses spiky representations and mitigates memorization on limited-data settings, making the practical takeaway more concrete.

Remaining concerns: A reviewer remains unconvinced about novelty and practical significance. In particular, they view the “spiky representation as memorization signature” as closely overlapping with prior work and consider the editing/steering component closer to existing latent manipulation techniques, with limited quantitative evaluation and only modest demonstrations on newer architectures. While the rebuttal improved clarity and added supportive experiments, this difference in perception about originality and impact is not fully resolved.

**Reviewer Scores:**

Eg8y: likely 6
NqY1: likely 6
jzL9: likely 8
6mi7: likely 4

---

### Decision · Program_Chairs · 2026-01-26

Accept (Poster)